# Improved inter-subject alignment of the lumbosacral cord for group-level in vivo gray and white matter assessments: A scan-rescan MRI study at 3T

**Silvan Büeler**[1], **Patrick Freund**[2,3,4], **Thomas M. Kessler**[1], **Martina D. Liechti**[1], **Gergely David**[1,2]*

1 Department of Neuro-Urology, Balgrist University Hospital, University of Zürich, Zürich, Switzerland,
2 Spinal Cord Injury Center, Balgrist University Hospital, University of Zürich, Zürich, Switzerland,
3 Department of Neurophysics, Max Planck Institute for Human Cognitive and Brain Sciences, Leipzig, Germany, 4 UCL Queen Square Institute of Neurology, Wellcome Trust Centre for Human Neuroimaging, University College London, London, United Kingdom

* gergely.david@balgrist.ch

**Data Availability Statement:** The data underlying the results presented in the study are available on

## Abstract

### Introduction

Magnetic resonance imaging (MRI) enables the investigation of pathological changes in gray and white matter at the lumbosacral enlargement (LSE) and conus medullaris (CM). However, conducting group-level analyses of MRI metrics in the lumbosacral spinal cord is challenging due to variability in CM length, lack of established image-based landmarks, and unknown scan-rescan reliability. This study aimed to improve inter-subject alignment of the lumbosacral cord to facilitate group-level analyses of MRI metrics. Additionally, we evaluated the scan-rescan reliability of MRI-based cross-sectional area (CSA) measurements and diffusion tensor imaging (DTI) metrics.

### Methods

Fifteen participants (10 healthy volunteers and 5 patients with spinal cord injury) underwent axial T2*-weighted and diffusion MRI at 3T. We assessed the reliability of spinal cord and gray matter-based landmarks for inter-subject alignment of the lumbosacral cord, the inter-subject variability of MRI metrics before and after adjusting for the CM length, the intra- and inter-rater reliability of CSA measurements, and the scan-rescan reliability of CSA measurements and DTI metrics.

### Results

The slice with the largest gray matter CSA as an LSE landmark exhibited the highest reliability, both within and across raters. Adjusting for the CM length greatly reduced the inter-subject variability of MRI metrics. The intra-rater, inter-rater, and scan-rescan reliability of MRI metrics were the highest at and around the LSE (scan-rescan coefficient of variation <3%

GitHub (https://github.com/NeuroimagingBalgrist/
LumbosacralCordMRI/tree/main/reliability).

**Funding:** This work is financially supported by the
Swiss National Science Foundation (SNSF)
(33IC30_179644). PF is funded by a SNSF
Eccellenza Professorial Fellowship grant
(PCEFP3_181362/1).

**Competing interests:** The authors have declared
that no competing interests exist.

for CSA measurements and <7% for DTI metrics within the white matter) and decreased
considerably caudal to it.

## Conclusions

To facilitate group-level analyses, we recommend using the slice with the largest gray mat-
ter CSA as a reliable LSE landmark, along with an adjustment for the CM length. We also
stress the significance of the anatomical location within the lumbosacral cord in relation to
the reliability of MRI metrics. The scan-rescan reliability values serve as valuable guides for
power and sample size calculations in future longitudinal studies.

## 1. Introduction

The lumbosacral spinal cord (SC) contains nuclei that innervate the lower limbs and pelvic
organs. Pathological changes in the gray matter (GM) or white matter (WM) of the lumbosa-
cral cord can lead to various dysfunctions such as motor and sensory impairments in the lower
limbs, as well as lower urinary tract, sexual and bowel dysfunction [1–3].

Pathological changes in the SC can be investigated in vivo by utilizing magnetic resonance
imaging (MRI) [4,5]. Cross-sectional area (CSA) measurements of SC, GM, and WM, derived
from multi-echo gradient-echo sequences, have been utilized as indirect measures of atrophy
in the cervical cord and lumbosacral enlargement (LSE) [6–12]. Furthermore, advanced MRI
techniques, such as diffusion MRI, have provided complementary information on WM integ-
rity [6–9,11,13–16]. In contrast, there have been only a few studies investigating the conus
medullaris (CM) extending caudally from the LSE [17,18]. Imaging the entire lumbosacral
cord is of particular interest, as tissue damage may also occur [19] or even originate within the
CM [20].

In the lumbosacral cord, SC level-specific and group-level analyses are challenged by the
mismatch between vertebral and neurological levels [21], which precludes the use of vertebral
levels as neuroanatomical landmarks. Given the difficulty of identifying neurological levels in
vivo in the lumbosacral cord [22], previous studies have suggested the use of the slice with the
largest SC CSA, here referred to as the "LSE landmark", as an image-based neuroanatomical
landmark [18,23]. However, the intra- and inter-rater reliability of image-based landmarks
have not been reported. Additionally, slice-wise group comparisons of MRI metrics are
increasingly challenging toward the tip of the spinal cord due to the high inter-subject variabil-
ity in the CM length [18].

Automatic SC and GM segmentation techniques have been shown to perform well for
the cervical cord [24] but have not yet been optimized for the lumbosacral cord. This is pri-
marily attributed to the smaller size and lower signal-to-noise ratio, as well as the close
proximity of nerve roots, which can compromise SC segmentation. Consequently, manual
segmentation is still the standard segmentation technique for the lumbosacral cord. For
healthy volunteers, the feasibility of manual GM and WM segmentation has been demon-
strated within the LSE [23] and CM [18], while the feasibility of diffusion tensor imaging
(DTI) has been shown within the LSE [25]. However, none of these studies reported scan-
rescan values for the CM.

The aim of this study is twofold. First, to facilitate group-level analyses, we aimed to
improve the inter-subject alignment of the lumbosacral cord by comparing GM- and SC-
based landmark definition methods and proposing a method to adjust for the individual CM

length. Second, we computed intra-rater, inter-rater, and scan-rescan reliability of the SC, GM, and WM CSA, as well as scan-rescan reliability of the DTI metrics, within both LSE and CM, and both in healthy volunteers and patients. Here, our focus was on measuring the reliability of MRI metrics in a patient cohort where the lumbosacral cord is not directly affected. For this reason, we included patients with a spinal cord injury (SCI) in the cervical or thoracic cord, who represent a challenging imaging cohort as they often experience higher levels of involuntary motion (e.g., due to spasticity).

## 2. Materials and methods

### 2.1 Study participants

Fifteen participants including 10 healthy volunteers (4 females, 6 males, age (mean ± standard deviation (SD)): 32.9 ± 10.8 years) and 5 SCI patients with cervical or thoracic injuries, but without injury-related radiological structural abnormalities in the lumbosacral cord (1 female, 4 males, age: 46.1 ± 20.3 years, time since injury: 5.9 ± 0.1 months) participated in this study. SCI patients had neurological impairments with mixed aetiologies, injury levels, and severities (see S1 Table for demographic and clinical information). The patient cohort was part of a randomized controlled trial investigating the effect of transcutaneous tibial nerve stimulation on the emergence of neurogenic lower urinary tract dysfunction following SCI (TASCI, ClinicalTrials.gov identifier: NCT03965299) [26]. The main inclusion criteria for all participants were: (i) no MRI contraindications, (ii) no pre-existing neurological or mental disorders, and (iii) > 18 years of age. For a comprehensive list of inclusion and exclusion criteria specific to SCI patients, refer to [26]. Healthy volunteers were scanned twice, with an interval of (mean ± SD) 8.4 ± 3.0 weeks (range: 6–15 weeks) between scans. Participants were recruited between 17.04.2020 and 14.02.2022. The study was approved by the local ethics committee (Kantonale Ethikkommission Zürich, BASEC ID: 2019–00074) and conducted in accordance with the Declaration of Helsinki. Written informed consent was obtained from all participants.

### 2.2 MRI acquisition

Scanning was performed on a 3T Siemens Prisma MRI scanner (Siemens Healthineers, Erlangen, Germany) using a body transmit coil and a 32-channel spine matrix coil. Foam wedges were placed beneath the knees to minimize the lower spine natural lordotic curve and maximize contact between the lower back and the spine matrix coil. Motion artifacts in the lower back area were reduced by placing the legs onto a spine vacuum cushion and applying Velcro straps around the knees, hips, and chest.

A sagittal T2-weighted turbo spin echo sequence was acquired as an anatomical reference of the lumbosacral cord as described in [17]. Subsequently, axial T2*-weighted images were acquired using a 3D spoiled multi-echo gradient-echo sequence (ME-GRE, Siemens FLASH) with 20 axial-oblique slices of 5 mm thickness (no gap) (Fig 1). A saturation band was placed anterior to the spine to suppress potential artifacts from the abdomen. The sequence parameters were as described in [17].

Images for diffusion MRI were acquired using a reduced field of view (FOV) single-shot spin-echo echo planar imaging (EPI) sequence with 15 slices of 5 mm thickness (no gap) (Fig 1). To prevent fold-over artifacts, two saturation bands were respectively placed anterior and posterior to the FOV. The sequence consisted of 180 diffusion-weighted (b = 800 s/mm$^2$) and 6 T2-weighted (b = 0 s/mm$^2$) images with an in-plane resolution of 0.9x0.9 mm$^2$, in-plane FOV of 86x32 mm$^2$, repetition time of 440 ms, echo time of 56 ms, no parallel imaging, 7/8 phase partial Fourier in the anterior-posterior phase-encoding direction, and bandwidth of

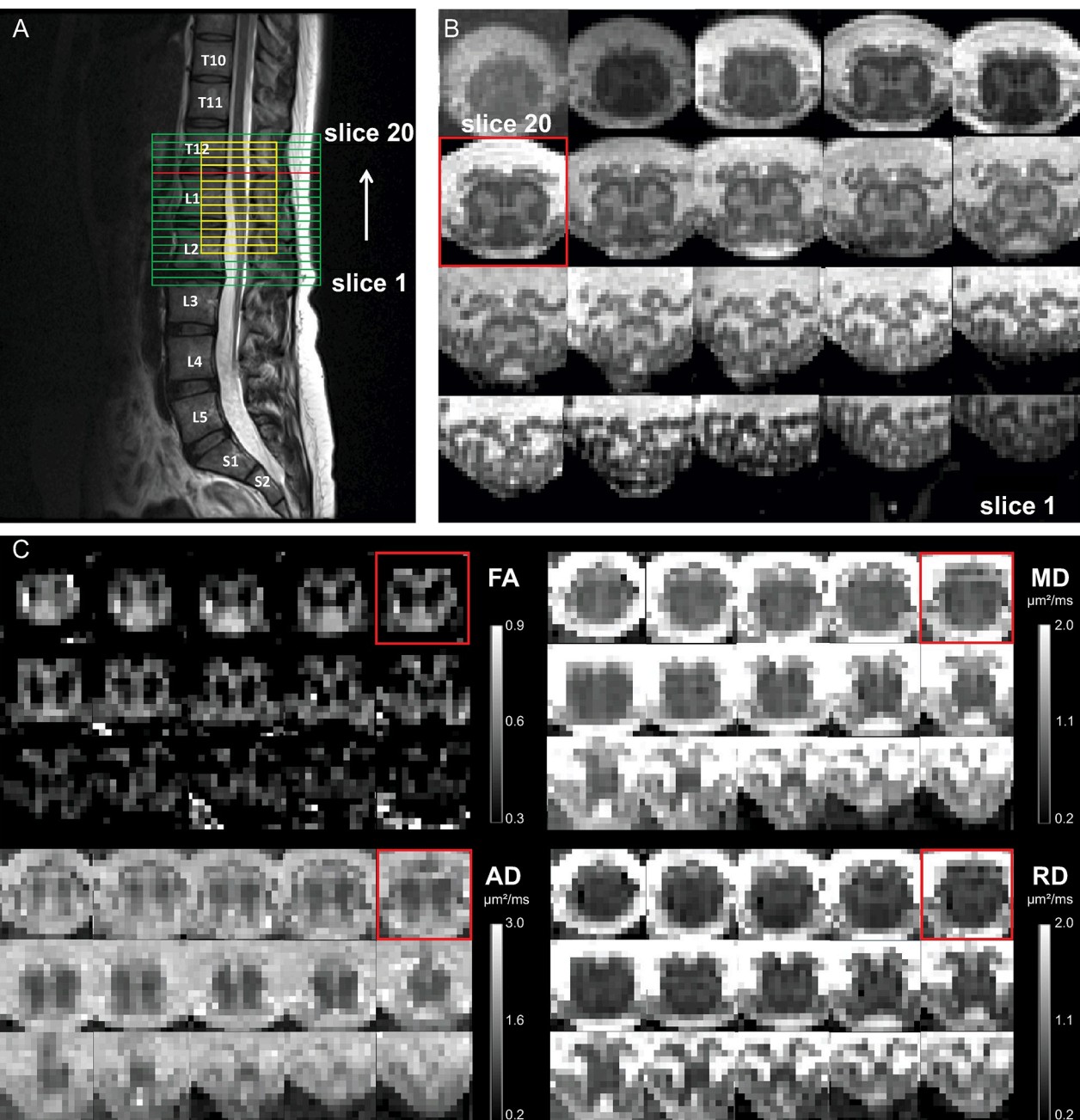

**Fig 1.** (A) The axial slice stacks of the 3D spoiled multi-echo gradient-echo (ME-GRE) sequence (indicated in green) and the diffusion MRI sequence (indicated in yellow), overlaid on the sagittal T2-weighted image. The field of view of the ME-GRE scan was set such that its 6th most rostral slice (slice #15) aligned with the maximum width of the spinal cord as observed in the sagittal T2-weighted scan (slice highlighted in red across all images). The slice stack of the ME-GRE scan covers the lumbosacral enlargement (LSE) and the entire conus medullaris (CM), while that of the diffusion MRI is smaller and centered at the 9th most rostral slice (slice #12) of the ME-GRE scan. (B) Corresponding axial slices of the ME-GRE scan. (C) Corresponding axial slices of the maps of diffusion tensor imaging metrics, including fractional anisotropy (FA), mean diffusivity (MD), axial diffusivity (AD), and radial diffusivity (RD). Axial slices are displayed in rostral (top left) to caudal (bottom right) direction.

1270 Hz/pixel. The acquisition was cardiac gated, acquiring 3 slices per cardiac cycle with a trigger delay of 120 ms. The total acquisition time depended on the heart rate and was approximately 12 min.

## 2.3 Processing of ME-GRE images

For each repetition, the first three echoes of the ME-GRE scan were combined via root-mean-squares, as it has been demonstrated to be the optimal combination for segmenting the SC and GM within the same image [17]. The resulting combined echo was then averaged across all repetitions. The SC and GM were manually segmented according to a standard operating procedure (SOP), made available on GitHub (https://github.com/NeuroimagingBalgrist/LumbosacralCordMRI), using the sub-voxel segmentation tool in JIM 7.0 (Xinapse systems), providing corresponding CSA values. WM CSA was obtained by subtracting GM CSA from SC CSA. Sub-voxel segmentations were binarized at an inclusion threshold of 100% for SC and 50% for GM to create binary SC and GM masks. A binary WM mask was created by subtracting the binary GM mask from the binary SC mask.

The three raters were instructed to segment the SC and GM by drawing an isointense contour within the partial volumes along the edges of the SC and GM, respectively, while also considering the anatomical shape and smoothness of the SC and GM. The raters had extensive experience in manually segmenting the lumbosacral cord, having individually segmented over 100 images prior to this study. To establish consensus guidelines for segmentation, the raters initially segmented a separate training set comprising three healthy volunteers, which was not included in the main analysis. These initial segmentations were then compared, and any disagreements were resolved through discussion among the raters. The resulting consensus guidelines were added to the SOP.

## 2.4 Processing of diffusion MRI images

The diffusion MRI images were processed using the ACID toolbox [27]. All images were cropped to an in-plane FOV of 32x32 mm$^2$. Eddy current and motion correction was performed using ECMOCO [28] with a 3-degrees-of-freedom (DOF) volume-wise registration (translation along x and y; scaling along y (with x and y being the left-right and anterior-posterior direction, respectively)), followed by a 2-DOF slice-wise registration (translation and scaling along y). Images underwent adaptive smoothing using msPOAS [29], with parameters k* = 5 and lambda = 10, to increase the signal-to-noise ratio without introducing blurring across tissue edges. The diffusion tensor model was then fitted on the corrected images using a robust tensor fitting algorithm [30,31] to generate maps of fractional anisotropy (FA), mean diffusivity (MD), axial diffusivity (AD), and radial diffusivity (RD).

The mean corrected diffusion-weighted image was manually segmented for SC in FSLeyes and was spatially normalized to the PAM50 template [32] resulting in both forward (native to template space) and backward (template to native space) warping fields. Normalization was aided by labeling both the slice with the largest SC CSA and the most caudal slice of the SC (determined from the ME-GRE sequence), which were aligned with the corresponding labels in the PAM50 template (disc labels 19 and 21). The probabilistic WM atlas, integrated into the PAM50 template, was warped into the native space using the obtained backward warping field. Using *sct_extract_metric* from the Spinal Cord Toolbox (v.5.8) [33], weighted average values of DTI metrics were extracted slice-wise within the GM, WM, as well as the dorsal, lateral, and ventral WM columns. As the PAM50 atlas does not encompass the caudal half of the CM, DTI metrics were extracted only from the upper half of the CM.

## 2.5 Comparison of neuroanatomical landmark definition methods

The LSE landmark as an image-based neuroanatomical landmark was defined either as the slice with the largest SC CSA or the largest GM CSA. The curves of slice-wise CSA values (see Fig 3 for examples) were either unsmoothed or underwent smoothing by moving window

averaging across three adjacent slices (corresponding to 15 mm). The choice of three slices was driven by the intrinsic smoothness of the curves of slice-wise CSA values and the need for selecting an uneven number of slices (thus avoiding the need for interpolation). Overall, this resulted in a total of four LSE landmark definition methods:

i.  $SC_{max}$: slice with the largest SC CSA without moving window averaging

ii.  $SC_{max,mw}$: slice with the largest SC CSA with moving window averaging

iii.  $GM_{max}$: slice with the largest GM CSA without moving window averaging

iv.  $GM_{max,mw}$: slice with largest GM CSA with moving window averaging

To assess the reliability of the LSE landmarks, the mean absolute deviation (MAD) of the determined slices was calculated across three sets of segmentations performed by the same rater (intra-rater reliability) and across segmentations performed by three different raters on the same set (inter-rater reliability) (see Section 2.7 for segmentation procedures).

Identifying the tip of the spinal cord (CMtip landmark) was difficult due to the large slice thickness (5 mm). Therefore, it was determined by extrapolating the curve of slice-wise SC CSA values to zero. Note that the most caudal slice where SC was segmented was usually 1–2 slices above the CMtip landmark.

## 2.6 Adjusting for the conus medullaris length

To adjust for the individual CM length, we divided the CM, i.e., space between the two landmarks (LSE and CMtip), into five segments of equal thickness in each subject. For example, if the CM consisted of 8 slices (40 mm) in length, each of the five segments comprised 8/5 = 1.6 slices (equivalent to 8 mm). Without reslicing the images, we extracted CSA values and DTI metrics within each segment, computed as a weighted average of the slice-wise values, where the weights represent the spatial contribution of each slice to the particular segment. This allowed us to determine MRI metrics within segments centered at the LSE and the CMtip landmarks, respectively, four segments located between them, and three segments rostral to the LSE landmark (Fig 2).

## 2.7 Intra- and inter-rater reliability of cross-sectional area measurements

A total of 10 participants including 5 healthy volunteers (1 female, 4 males, age (mean ± SD): 41.4 ± 8.6 years) and 5 patients with SCI (1 female, 4 males, age: 46.1 ± 20.3 years) were included in this analysis. No images had to be discarded due to motion or other artifacts. Segmentation of the axial ME-GRE images was performed by three raters (S.B., G.D., M.D.L.). Two raters performed the segmentation three times and one rater performed it once, with a minimum of two weeks between each round of segmentation. The order of segmentation in each round was pseudo-randomized into three blocks using a computer-generated randomization list to ensure a balanced distribution of patients and healthy volunteers. Raters were provided with the pseudonymized set of images (ME-GRE of the lumbosacral cord). The raters were not provided with images covering the injury site, where the participant status would have been visible. Since the patients did not exhibit any injury-related radiological abnormalities or implant-related artifacts in the lumbosacral cord, raters were unable to discern whether the participant was a healthy volunteer or a patient. The LSE landmark, defined as the $GM_{max,mw}$ slice, was determined in each subject as the median across three raters.

Slice-wise coefficient of variation (CV) of SC, GM, and WM CSA was calculated as the percent ratio between the standard deviation and mean of CSA values either (i) across three sets of segmentations performed by the same rater (intra-rater reliability) or (ii) across

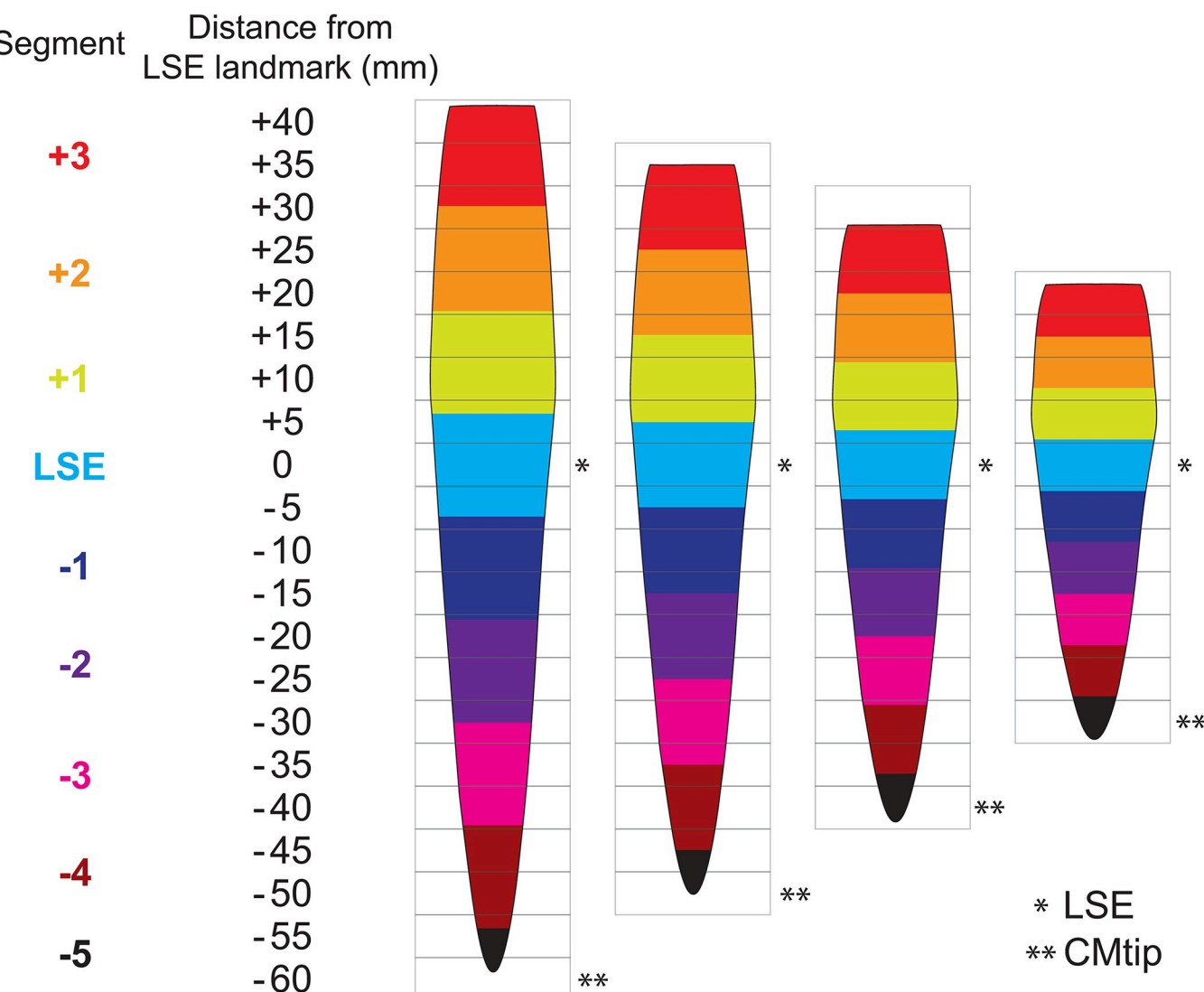

**Fig 2. Adjustment for the individual conus medullaris (CM) length (60, 50, 40, and 30 mm for the displayed cases).** The acquired slices are illustrated by rectangular gray boxes (slice thickness of 5 mm). First, two image-based neuroanatomical landmarks (LSE, defined here as the slice with the largest cross-sectional GM area, and the CMtip) are determined in each subject (indicated by asterisks). Then, the space between these landmarks are divided into 5 segments of equal thickness (resulting in segment thicknesses of 12, 10, 8, and 6 mm). The segments, displayed as colored spinal cord regions, are defined such that one segment (light blue) is centered at the LSE landmark (segment LSE), another one at the tip of the spinal cord (segment LSE-5), and the space between them is covered by four segments. The spinal cord rostral to the LSE landmark is also divided into segments using the same segment thickness. Average values are extracted within each segment as a weighted average of the slice-wise values, where the weights represent the spatial contribution of each slice to the segment. If a value for a slice which contributes more than 25% to the segment was not available, the value for that segment was not calculated.

segmentations performed by three different raters on the same set (inter-rater reliability). The CV was then averaged across all subjects (n = 10) and separately for patients (n = 5) and healthy volunteers (n = 5).

Intra- and inter-rater intraclass correlation coefficients (ICC) were simultaneously computed using Eliasziw's two-way mixed effects, absolute agreement, single-measure model [34], as implemented in the *relInterIntra* function of the *irr* package in R. Two-sided (upper and lower bounds) 95% confidence intervals were additionally calculated according to the formulas in [35]. The Dice coefficient represents the spatial overlap between masks and was computed as the size of the union of two (binary) segmentation masks, created either by the same rater

(intra-rater Dice coefficient) or different raters (inter-rater Dice coefficient), divided by the average size of the two segmentations masks. Values range from 0 (no overlap) to 1 (perfect overlap). Single Dice coefficients were calculated by averaging pairwise Dice coefficients for each subject, and then further averaged across all subjects.

## 2.8 Scan-rescan reliability of cross-sectional area measurements and diffusion tensor imaging

The reliability analysis across both imaging sessions (scan and rescan) included the 10 healthy volunteers. Patients were not included, as their scan-rescan data might be affected by disease-related longitudinal changes. No images had to be discarded due to motion or other artifacts. Segmentation was performed by a single rater (S.B.). The LSE landmark was defined as the $GM_{max,mw}$ slice.

We calculated the mean of the scan-rescan differences ($\bar{d}$), along with the 95% limits of agreement ($\bar{d} \pm 1.96 \cdot SD$). Scan-rescan CV was calculated as the percent ratio between the standard deviation and mean of MRI metrics across scan and rescan. Scan-rescan ICC was computed using a two-way mixed effects, absolute agreement, single-measure model using the *icc* function of the *irr* package in R. The minimal detectable change (MDC), or smallest real difference represents the smallest longitudinal change that be confidently attributed to a meaningful change rather than mere scan-rescan variability. The MDC with 95% confidence threshold were calculated as $MDC = 1.96 \cdot \sqrt{2} \cdot SD_{total} \cdot \sqrt{1 - ICC}$, where $SD_{total}$ denotes the total standard deviation (taken across all measurements) and ICC denotes the scan-rescan ICC. While MDC describes the minimal detectable longitudinal change in an individual, it is important to note that group studies can detect even smaller longitudinal changes by utilizing multiple subjects.

## 3. Results

### 3.1 Comparison of neuroanatomical landmark definition methods

Visual comparison of different LSE landmarks is provided in Fig 3. The inter-rater MAD values of the LSE landmarks were 0.58 for $SC_{max}$, 0.42 for $SC_{max,mw}$, 0.29 for $GM_{max}$, and 0.22 for $GM_{max,mw}$. Intra-rater MAD values were consistently the same or lower than corresponding inter-rater values (0.16 for $SC_{max}$, 0.19 for $SC_{max,mw}$, 0.29 for $GM_{max}$, and 0.11 for $GM_{max,mw}$). The lowest intra- and inter-rater MAD values were achieved for $GM_{max,mw}$. The $GM_{max,mw}$ slice was located (mean ± SD) 8.5 ± 4.5 mm more caudally than the $SC_{max,mw}$ slice. The inter-subject variability of SC and GM CSA was lower at caudal locations when using the $GM_{max,mw}$ slice as LSE landmark; however, no clear trend was observed at rostral locations (S2 Table, Fig 4).

### 3.2 Adjusting for the conus medullaris length

After adjusting for the CM length, the inter-subject CV of CSA measurements remained similar at and around the $GM_{max,mw}$ slice, but decreased substantially caudal to it (Table 1). For example, the inter-subject CV of SC CSA was 31.5% five slices (25 mm) below the $GM_{max,mw}$ slice (before adjustment), and it reduced to 17.4% after adjustment at roughly the same anatomical location (segment LSE-3) (compare values highlighted in bold in Table 1).

### 3.3 Intra- and inter-rater reliability of cross-sectional area measurements

Fig 5 illustrates the intra- and inter-rater variability in SC and GM segmentations, while Table 2 lists the intra- and inter-rater reliability metrics for the CSA measurements. Intra-rater reliability was higher than inter-rater reliability for all CSA values and slices, indicated by lower CV and higher ICC and Dice coefficients. Intra- and inter-rater reliability was in general

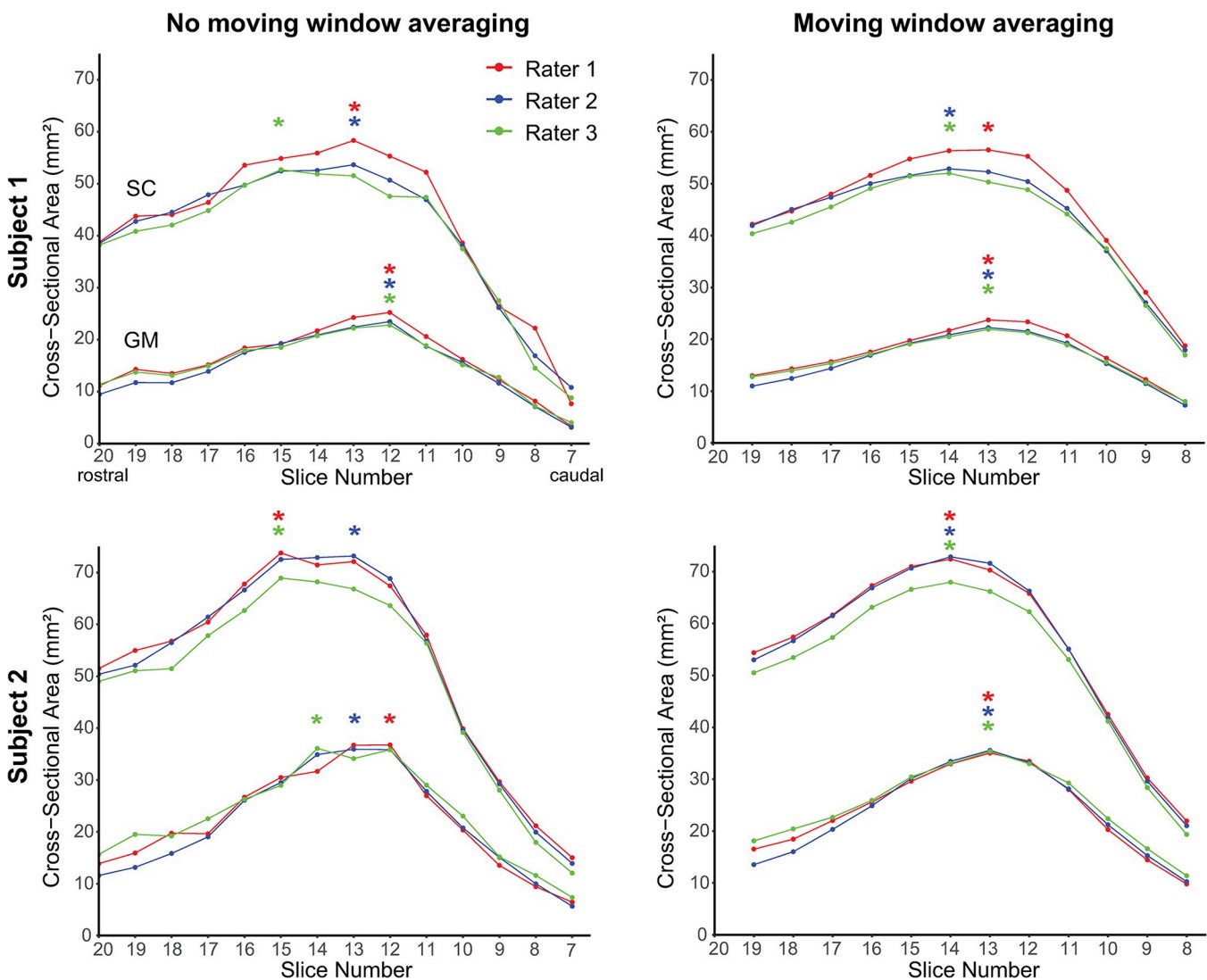

**Fig 3. Curves of slice-wise cross-sectional area (CSA) of the spinal cord (SC) and gray matter (GM), obtained by three different raters in two subjects, with and without applying moving window averaging across 3 adjacent slices.** The LSE landmarks, as determined by the raters, are indicated by asterisks. The benefit of using the slice with the largest GM CSA ($GM_{max,mw}$) as LSE landmark, as opposed to the slice with the largest SC CSA ($SC_{max,mw}$), is evident in Subject 1, with no inter-rater variability for the $GM_{max,mw}$ slice. Subject 2 demonstrates the advantage of using moving window averaging: There is no inter-rater variability in $GM_{max,mw}$ and $SC_{max,mw}$ slice after applying moving window averaging.

higher for SC CSA than for GM and WM CSA. Intra- and inter-rater CV increased considerably from the $GM_{max,mw}$ slice toward the tip of the spinal cord, accompanied by a decrease in Dice coefficients. Notably, such a trend could not be observed in the ICC. Intra-rater CV values were similar between healthy volunteers and patients, but inter-rater CV values were higher in the patient group, especially at caudal locations (S3 Table).

### 3.4 Scan-rescan reliability of cross-sectional area measurements and diffusion tensor imaging

There was no systematic bias, as measured by $\bar{d}$, between the scan and rescan values for any of the MRI metrics (Tables 3–5, S1 Fig). In general, scan-rescan reliability was higher for SC CSA than for GM and WM CSA, indicated by lower CV and MDC, and higher ICC values

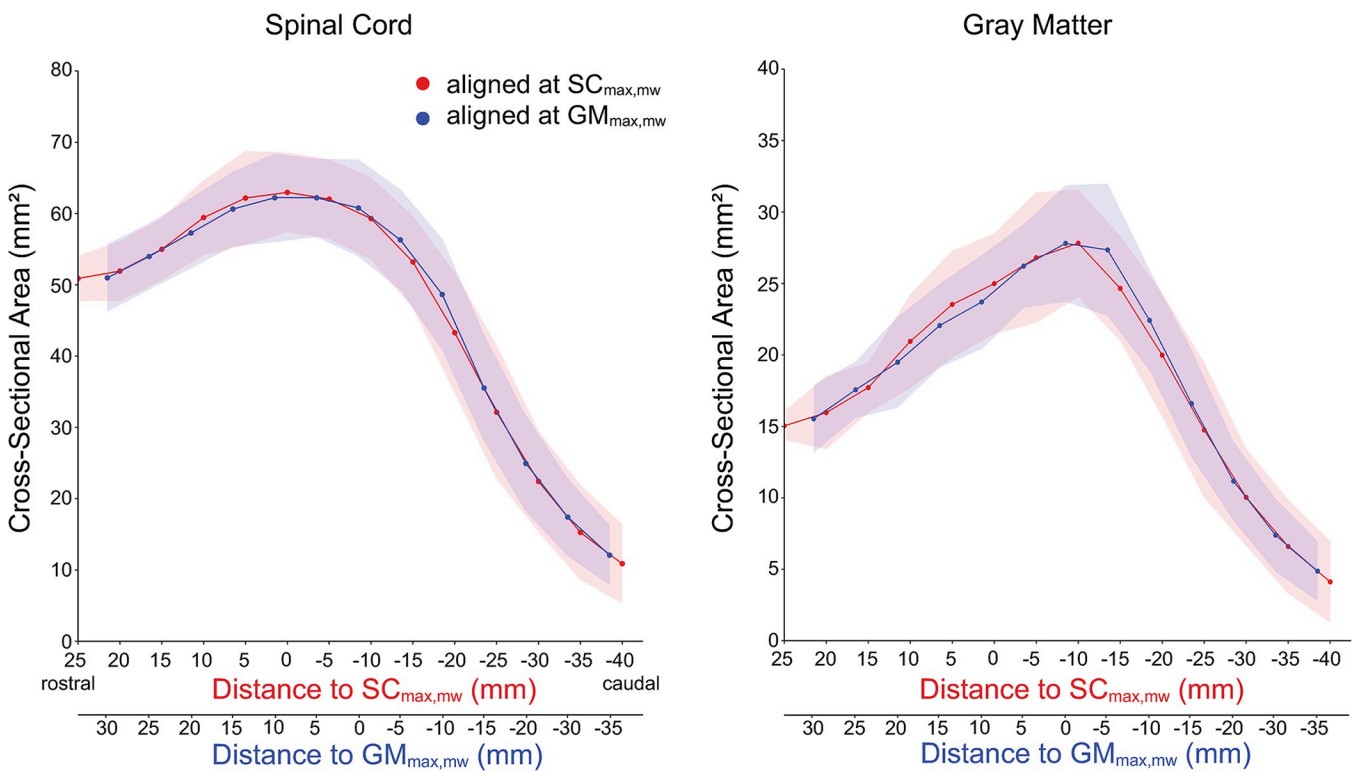

**Fig 4. Inter-subject mean (solid line) and standard deviation (shaded area) of the cross-sectional area (CSA) of the spinal cord (SC) and gray matter (GM), computed across 10 healthy volunteers, when aligning the individual slice stacks at the slice either with the largest SC CSA (SC$_{max,mw}$, in red) or GM SCA (GM$_{max,mw}$, in blue), without adjusting for the length of the conus medullaris.** The inter-subject variability at caudal locations was slightly lower when aligning at the GM$_{max,mw}$ slice, as seen by the smaller width of the blue shaded areas.

(Table 3). Among the DTI metrics, RD tended to have the lowest scan-rescan reliability. In general, reliability was higher for DTI metrics when extracted within the entire WM rather than separate WM columns. Reliability (with the exception of ICC values for DTI metrics) decreased considerably from the LSE segment toward the tip of the SC. Without adjusting for the CM length, the scan-rescan CV values for DTI metrics were higher (compare Tables 3–5 with S4–S6 Tables, respectively). The scan-rescan reliability decreased only minimally (i.e., the increase in CV was, on average, below 1 percentage point for MRI metrics extracted within the WM) when the landmarks were determined independently for scan and rescan, as opposed to when they were determined in the first scan (compare Tables 3–5 with S7–S9 Tables, respectively).

## 4. Discussion

We investigated methods for improving the inter-subject alignment of axial slice stacks within the lumbosacral cord. We found that the slice with the largest gray matter (GM) cross-sectional area (CSA) can serve as a reliable image-based neuroanatomical landmark for the lumbosacral enlargement (LSE). Adjusting for the conus medullaris (CM) length by dividing the CM into a fixed number of segments and extracting MRI metrics from those segments substantially reduced inter-subject variability. Additionally, we reported different aspects of reliability for CSA measurements and diffusion tensor imaging (DTI) metrics within the lumbosacral cord at 3T. We found that intra-rater, inter-rater, and scan-rescan reliability was the highest at and around the LSE landmark, and decreased caudal to it.

**Table 1. Inter-subject variability of cross-sectional area measurements.**

| | | Spinal Cord | | Gray Matter | | White Matter | |
|---|---|---|---|---|---|---|---|
| | | CSA (mm$^2$) | CV (%) | CSA (mm$^2$) | CV (%) | CSA (mm$^2$) | CV (%) |
| Distance from LSE landmark (mm) | +15 | 60.6 ± 5.3 | 8.7 | 22.1 ± 2.9 | 13.3 | 38.6 ± 3.3 | 8.4 |
| | +10 | 62.2 ± 6.2 | 10.0 | 23.7 ± 3.3 | 13.8 | 38.5 ± 3.8 | 9.8 |
| | +5 | 62.2 ± 5.5 | 8.8 | 26.2 ± 2.9 | 11.2 | 36.0 ± 3.1 | 8.6 |
| | 0 | 60.8 ± 6.8 | 11.2 | 27.8 ± 4.1 | 14.7 | 33.0 ± 4.3 | 13.2 |
| | -5 | 56.3 ± 7.1 | 12.5 | 27.3 ± 4.6 | 16.9 | 29.0 ± 3.2 | 10.9 |
| | -10 | 48.7 ± 7.8 | 16.1 | 22.4 ± 3.5 | 15.7 | 26.2 ± 4.5 | 17.1 |
| | -15 | 35.5 ± 7.5 | 21.2 | 16.6 ± 3.8 | 22.6 | 18.9 ± 4.1 | 21.5 |
| | -20 | 24.9 ± 6.8 | 27.2 | 11.2 ± 2.8 | 24.9 | 13.8 ± 4.2 | 30.2 |
| | -25 | **17.4 ± 5.5** | **31.5** | **7.4 ± 2.6** | **34.6** | **10.0 ± 3.1** | **30.6** |
| Segment (adjusted) | +2 | 59.4 ± 5.7 | 9.6 | 21.5 ± 3.3 | 15.3 | 38.0 ± 3.0 | 7.9 |
| | +1 | 62.0 ± 5.7 | 9.1 | 24.6 ± 3.2 | 12.9 | 37.3 ± 3.5 | 9.3 |
| | LSE | 60.3 ± 6.6 | 10.9 | 27.5 ± 3.9 | 14.2 | 32.8 ± 3.7 | 11.2 |
| | -1 | 51.6 ± 7.2 | 14.0 | 24.5 ± 4.1 | 16.6 | 27.1 ± 3.5 | 12.7 |
| | -2 | 34.2 ± 6.0 | 17.5 | 15.9 ± 3.4 | 21.1 | 18.3 ± 2.9 | 15.7 |
| | -3 | **18.7 ± 3.3** | **17.4** | **8.3 ± 1.7** | **19.8** | **10.3 ± 1.7** | **16.3** |
| | -4 | 9.9 ± 1.7 | 17.4 | 3.9 ± 0.5 | 13.2 | 6.0 ± 1.3 | 21.9 |

*Notes*: CSA values represent mean ± standard deviation across 10 healthy volunteers. Individual axial slice stacks were aligned at the LSE landmark, defined as the slice with the largest gray matter CSA (GM$_{max,mw}$). Values are displayed for both original slices (not adjusted for the length of the conus medullaris) and segments (adjusted for the length of the conus medullaris). While a direct correspondence between the original slices and segments is not possible due to their different thickness, the rows highlighted in bold roughly represent the same anatomical location. A positive distance indicates a rostral direction from the LSE landmark. Values were derived from the first scan; highly comparable results were obtained from the second scan (rescan).

*Abbreviations*: CV, inter-subject coefficient of variation; CSA, cross-sectional area; LSE, lumbosacral enlargement.

## 4.1 Slice with the largest gray matter cross-sectional area is a reliable image-based neuroanatomical landmark

We found that the slice with the largest GM CSA, as opposed to the slice with the largest spinal cord (SC) CSA, can be identified more consistently across raters and across repeated assessments of the same rater. This is likely because of the sharper peak in the curves of slice-wise GM CSA values at the LSE compared to the flatter peak in SC CSA (Fig 3). Sharper peaks are easier for raters to identify than flatter peaks. Further improvement in the reliability of image-based landmarks can be achieved by smoothing the curves of slice-wise CSA values through moving window averaging across three adjacent slices, which reduces fluctuations inherent in the slice-wise CSA values.

Furthermore, when aligning the individual axial slice stacks at the LSE landmark determined based on GM CSA, as opposed to SC CSA, we noticed a slight reduction in inter-subject variability of CSA values at caudal locations. The reason for this reduction remains unclear. We argue that the slice with the largest GM CSA is more closely associated with the neurological level than the slice with the largest SC CSA, considering that the LSE is neuroanatomically attributed to the enlargement of the GM.

Notably, a previous study based on post-mortem MRI utilized distinct morphological features of the ventral GM horns to characterize the lumbosacral cord across species [36]. Another study revealed a strong link between these morphological features and the motor neuron pools located within the ventral GM horns [37]. We anticipate that, for in vivo studies, the shape of GM would serve as an even better neuroanatomical landmark than its size. In a recent

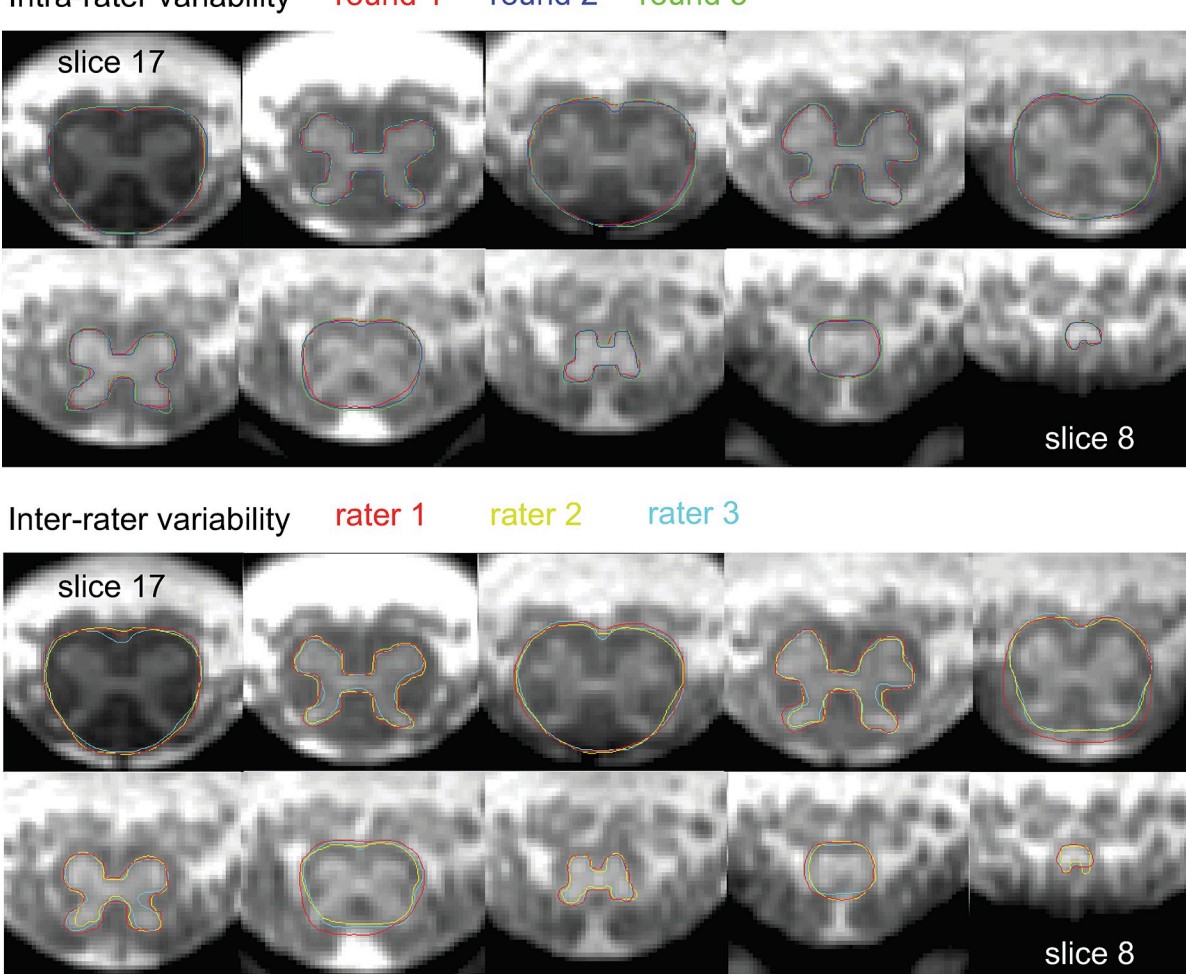

**Fig 5. Illustration of intra- and inter-rater variability of manual spinal cord (SC) and gray matter (GM) segmentations within a representative subject, i.e., a subject whose intra- and inter-rater variability values are close to the mean values reported in Table 2.** SC and GM segmentations are shown on alternating slices for display purposes. For the displayed SC and GM segmentations, the coefficient of variation (CV) values of the corresponding SC cross-sectional areas (CSA) were (intra- vs. inter-rater) 0.8% vs. 3.7% (slice 17), 2.4% vs. 3.2%, 1.2% vs. 9.9%, 4.8% vs. 12.3%, and 6.8% vs. 6.7% (slice 9). The CV values of the corresponding GM CSA were (intra- vs. inter-rater) 1.8% vs. 9.1% (slice 16), 1.5% vs. 9.1%, 2.4% vs. 8.9%, 6.5% vs. 10.9%, and 3.6% vs. 9.5% (slice 8).

study, a distinct approach was used to directly identify neurological levels through nerve root tracing [38]. However, this approach requires the acquisition of an additional image optimized for nerve segmentation, thereby increasing the scan time, and might not work reliably in all subjects. Future research should focus on identifying neurological levels directly from the axial ME-GRE images.

## 4.2 Improved inter-subject alignment after adjusting for the conus medullaris length

To adjust for the considerable inter-subject variation in CM length, we divided the CM into a fixed number of segments and extracted MRI metrics from these segments using a weighted average of the slice-wise values. This adjustment substantially reduced the inter-subject variability of MRI metrics at caudal locations. This is beneficial for group-level analyses, as

**Table 2. Slice-wise intra- and inter-rater reliability of cross-sectional area measurements (n = 10; 5 healthy volunteers and 5 patients with spinal cord injury).**

| | Distance from LSE landmark (mm) | CSA (mm²) mean ± SD | CV (%) | | ICC [95% CI] | | Dice coefficient | |
|---|---|---|---|---|---|---|---|---|
| | | | Intra-rater | Inter-rater | Intra-rater | Inter-rater | Intra-rater | Inter-rater |
| Spinal Cord | +20 | 53.7 ± 3.8 | 2.4 | 5.0 | .89 [.69, .95] | .62 [.19, .88] | 0.96 | 0.96 |
| | +15 | 57.2 ± 4.5 | 1.9 | 4.5 | .93 [.79, .97] | .72 [.29, .92] | 0.96 | 0.96 |
| | +10 | 59.4 ± 5.5 | 2.1 | 4.1 | .93 [.80, .97] | .80 [.31, .95] | 0.96 | 0.96 |
| | +5 | 59.1 ± 5.6 | 2.3 | 3.4 | .92 [.79, .97] | .88 [.38, .97] | 0.97 | 0.96 |
| | 0 | 58.5 ± 5.4 | 2.1 | 5.0 | .94 [.83, .98] | .76 [.17, .94] | 0.97 | 0.96 |
| | -5 | 54.8 ± 4.9 | 2.4 | 6.0 | .93 [.79, .97] | .67 [.12, .91] | 0.96 | 0.95 |
| | -10 | 46.5 ± 4.8 | 2.8 | 9.0 | .93 [.80, .97] | . 50 [.12, .82] | 0.95 | 0.94 |
| | -15 | 35.0 ± 4.5 | 3.8 | 9.1 | .95 [.82, .97] | .56 [.20, .84] | 0.95 | 0.95 |
| | -20 | 24.3 ± 4.8 | 4.0 | 9.3 | .95 [.86, .98] | .76 [.45, .93] | 0.94 | 0.94 |
| | -25 | 17.0 ± 4.5 | 5.4 | 13.3 | .94 [.83, .98] | .70 [.32, .91] | 0.93 | 0.92 |
| | -30 | 10.9 ± 4.4 | 7.8 | 16.2 | .96 [.88, .98] | .85 [.63, .96] | 0.91 | 0.90 |
| Gray Matter | +20 | 17.2 ± 2.3 | 3.9 | 7.5 | .88 [.66, .95] | .65 [.28, .89] | 0.94 | 0.92 |
| | +15 | 20.3 ± 2.6 | 4.8 | 6.0 | .81 [.53, .92] | .77 [.46, .93] | 0.94 | 0.92 |
| | +10 | 22.1 ± 3.4 | 3.9 | 6.2 | .90 [.73, .96] | .82 [.52, .95] | 0.95 | 0.93 |
| | +5 | 24.2 ± 4.1 | 3.6 | 4.9 | .92 [.78, .97] | .90 [.73, .97] | 0.95 | 0.93 |
| | 0 | 25.4 ± 4.3 | 3.6 | 6.8 | .94 [.84, .98] | .84 [.31, .96] | 0.95 | 0.92 |
| | -5 | 25.0 ± 4.4 | 2.7 | 7.3 | .96 [.89, .99] | .81 [.33, .95] | 0.95 | 0.93 |
| | -10 | 20.7 ± 3.1 | 3.9 | 10.5 | .92 [.79, .97] | .62 [.23, .87] | 0.94 | 0.92 |
| | -15 | 15.3 ± 3.1 | 4.0 | 11.9 | .95 [.85, .98] | .67 [.33, .89] | 0.94 | 0.91 |
| | -20 | 10.8 ± 2.4 | 6.0 | 13.0 | .93 [.79, .97] | .68 [.35, .90] | 0.93 | 0.91 |
| | -25 | 7.4 ± 2.3 | 7.3 | 14.1 | .94 [.82, .97] | .75 [.46, .92] | 0.91 | 0.88 |
| | -30 | 4.3 ± 2.2 | 10.8 | 13.9 | .94 [.82, .98] | .91 [.76, .98] | 0.90 | 0.88 |
| White Matter | +20 | 36.0 ± 2.2 | 3.0 | 6.9 | .82 [.50, .91] | .36 [.01, .75] | 0.91 | 0.90 |
| | +15 | 36.9 ± 2.2 | 3.2 | 6.3 | .78 [.44, .90] | .38 [.03, .75] | 0.91 | 0.90 |
| | +10 | 37.3 ± 2.4 | 3.7 | 5.8 | .79 [.49, .91] | .45 [.09, .79] | 0.91 | 0.90 |
| | +5 | 34.9 ± 2.3 | 3.6 | 4.6 | .69 [.32, .86] | .55 [.17, .84] | 0.91 | 0.89 |
| | 0 | 33.1 ± 1.6 | 3.0 | 5.0 | .74 [.41, .89] | .39 [.04, .76] | 0.90 | 0.88 |
| | -5 | 29.8 ± 2.2 | 3.4 | 6.1 | .80 [.41, .89] | .50 [.13, .82] | 0.89 | 0.86 |
| | -10 | 25.8 ± 2.6 | 5.1 | 10.5 | .80 [.46, .90] | .38 [.03, .75] | 0.87 | 0.85 |
| | -15 | 19.6 ± 2.4 | 5.7 | 11.1 | .87 [.60, .93] | .40 [.02, .77] | 0.85 | 0.85 |
| | -20 | 13.5 ± 2.7 | 6.1 | 9.9 | .91 [.72, .96] | .73 [.43, .92] | 0.84 | 0.83 |
| | -25 | 9.6 ± 2.4 | 7.5 | 19.2 | .88 [.63, .94] | .51 [.12, .82] | 0.82 | 0.78 |
| | -30 | 6.6 ± 2.3 | 9.8 | 22.4 | .92 [.76, .97] | .68 [.34, .91] | 0.76 | 0.78 |

Notes: CSA values represent an average across values obtained by three raters, based on the first set of segmentation of each rater. The individual axial slice stacks were aligned at the LSE landmark, defined as the slice with the largest gray matter CSA ($GM_{max,mw}$), without adjustment for the length of the conus medullaris. A positive distance indicates a rostral direction from the LSE landmark. For a single subject, GM and WM CSA values were not available for slices with coordinates +20 and -30 mm (n = 9).

*Abbreviations*: CI, confidence interval; CV, coefficient of variation; CSA, cross-sectional area; ICC, intraclass correlation coefficient; LSE, lumbosacral enlargement; SD, standard deviation.

reduced inter-subject variability increases statistical power for the same sample size or necessitates a smaller sample size for the same statistical power. We note that another line of research is concerned with reducing inter-subject variability of MRI metrics that arise from biological variability, using regression models incorporating demographic information, spine, and SC metrics [18,39]; however, this was not the focus of our investigation.

**Table 3. Scan-rescan reliability of cross-sectional area measurements (n = 10 healthy volunteers).**

| | Segment | CSA (mm²) mean ± SD | $\bar{d}$ (mm²) [± 1.96 SD] | CV (%) | ICC [95% CI] | MDC (%) |
|---|---|---|---|---|---|---|
| Spinal Cord | +3 | 55.2 ± 5.8 | -0.8 [±4.7] | 2.6 | .92 [.72, .98] | 8.4 |
| | +2 | 59.7 ± 5.8 | -0.4 [±3.4] | 1.6 | .96 [.85, .99] | 5.4 |
| | +1 | 61.8 ± 5.6 | 0.3 [±1.6] | 0.8 | .99 [.96, 1.00] | 2.6 |
| | LSE | 59.9 ± 6.7 | 0.8 [±3.0] | 1.8 | .97 [.87, .99] | 5.3 |
| | -1 | 50.7 ± 7.0 | 1.8 [±6.0] | 3.9 | .89 [.59, .97] | 12.9 |
| | -2 | 33.6 ± 5.8 | 1.2 [±6.5] | 6.3 | .84 [.52, .96] | 19.3 |
| | -3 | 18.4 ± 2.6 | 0.5 [±6.3] | 10.0 | .47 [-.22, .84] | 32.7 |
| | -4 | 9.5 ± 1.4 | 0.8 [±5.0] | 15.4 | .09 [-.54, .65] | 52.2 |
| | -5 | 5.9 ± 1.6 | 0.2 [±2.6] | 11.4 | .72 [.20, .92] | 41.1 |
| Gray Matter | +3 | 18.0 ± 3.4 | 0.1 [±2.2] | 3.7 | .95 [.82, .99] | 11.6 |
| | +2 | 21.7 ± 3.2 | -0.4 [±2.4] | 3.4 | .93 [.75, .98] | 11.1 |
| | +1 | 24.8 ± 3.2 | -0.2 [±2.4] | 3.2 | .93 [.77, .98] | 9.0 |
| | LSE | 27.2 ± 3.7 | 0.7 [±2.1] | 2.9 | .95 [.74, .99] | 8.7 |
| | -1 | 23.7 ± 3.9 | 1.5 [±3.6]* | 5.9 | .85 [.38, .96] | 18.0 |
| | -2 | 15.7 ± 3.1 | 0.5 [±3.8] | 8.3 | .83 [.48, .96] | 23.0 |
| | -3 | 8.3 ± 1.4 | 0.1 [±2.5] | 9.4 | .68 [.11, .91] | 28.8 |
| | -4 | 3.7 ± 0.6 | 0.2 [±2.0] | 13.3 | .15 [-.53, .69] | 52.4 |
| | -5 | 1.9 ± 0.6 | -0.1 [±1.4] | 18.5 | .55 [-.13, .87] | 69.5 |
| White Matter | +3 | 37.2 ± 2.9 | -0.9 [±5.3] | 4.5 | .64 [.09, .89] | 14.0 |
| | +2 | 38.0 ± 3.1 | 0.0 [±2.0] | 1.6 | .95 [.82, .99] | 4.9 |
| | +1 | 37.1 ± 3.2 | 0.5 [±1.7] | 1.4 | .95 [.80, .99] | 5.1 |
| | LSE | 32.7 ± 3.6 | 0.1 [±2.9] | 2.5 | .92 [.73, .98] | 8.4 |
| | -1 | 26.9 ± 3.3 | 0.4 [±4.4] | 4.8 | .80 [.39, .95] | 15.6 |
| | -2 | 17.9 ± 2.9 | 0.7 [±3.4] | 5.7 | .82 [.47, .95] | 19.1 |
| | -3 | 10.2 ± 1.3 | 0.4 [±3.9] | 10.6 | .24 [-.47, .74] | 37.2 |
| | -4 | 5.8 ± 0.9 | 0.5 [±3.2] | 17.1 | .08 [-.53, .65] | 55.5 |
| | -5 | 4.0 ± 1.1 | 0.3 [±1.6] | 11.2 | .74 [.28, .93] | 39.8 |

* Indicates significant difference between scan and rescan (p < 0.05).

*Notes*: The individual axial slice stacks were aligned at the LSE landmark, defined as the slice with the largest gray matter CSA ($GM_{max,mw}$), and were adjusted for the length of the conus medullaris. The landmarks were determined in the first scan. A positive segment indicates a rostral direction from the LSE landmark.

*Abbreviations*: CI, confidence interval; CV, scan-rescan coefficient of variation; CSA, cross-sectional area; $\bar{d}$, mean scan-rescan difference; ICC, scan-rescan intraclass correlation coefficient; LSE, lumbosacral enlargement; MDC, minimal detectable change; SD, standard deviation.

The decision of dividing the CM into five segments was determined by both the slice thickness (5 mm) and the range of CM lengths observed within the healthy population [18]. By using five segments, we ensured that segments were thicker than the slice thickness in all participants, avoiding the need for interpolating within slices. While thinner slices would allow for a higher number of segments and increased spatial specificity, this comes at the cost of reduced scan-rescan reliability. Conversely, dividing the CM into fewer segments would likely enhance scan-rescan reliability, but it would come at the expense of specificity along the rostro-caudal axis.

Spinal cord segments, created by correcting for the individual CM length, do not correspond to specific neurological levels. Instead, each segment consists of a combination of neurological levels, where more caudal segments encompass more neurological levels due to their progressively shorter length [40]. Nevertheless, this approach is still consistent with

**Table 4. Scan-rescan reliability of fractional anisotropy and mean diffusivity values (n = 10 healthy volunteers).**

| | | Fractional Anisotropy | | | | | Mean Diffusivity ($10^{-3}$ mm²/s) | | | | |
|---|---|---|---|---|---|---|---|---|---|---|---|
| | Segment | mean ± SD | $\bar{d}$ [± 1.96 SD] | CV (%) | ICC [95% CI] | MDC (%) | mean ± SD | $\bar{d}$ [± 1.96 SD] | CV (%) | ICC [95% CI] | MDC (%) |
| Gray Matter | +3 | .44 ± .07 | .00 [±.10] | 6.5 | .76 [.28, .94] | 21.2 | .90 ± .05 | .02 [±.09] | 3.0 | .62 [.08, .89] | 9.8 |
| | +2 | .38 ± .06 | .01 [±.11] | 7.0 | .70 [.18, 92] | 27.1 | .88 ± .05 | .01 [±.14] | 4.6 | .44 [-.27, .83] | 14.8 |
| | +1 | .34 ± .05 | -.01 [±.08] | 7.2 | .73 [.26, .92] | 23.7 | .86 ± .04 | .01 [±.09] | 2.8 | .45 [-.22, .83] | 10.0 |
| | LSE | .33 ± .04 | .00 [±.05] | 3.9 | .86 [.53, .96] | 13.4 | .81 ± .05 | .00 [±.13] | 5.1 | .38 [-.36, .80] | 15.5 |
| | -1 | .32 ± .06 | -.02 [±.05] | 5.3 | .87 [.52, .97] | 18.3 | .81 ± .03 | .00 [±.16] | 6.1 | -.44 [-.97, .31] | 20.1 |
| | -2 | .32 ± .07 | .00 [±.11] | 7.8 | .73 [.21, .93] | 34.1 | .81 ± .03 | -.01 [±.10] | 3.5 | .21 [-.52, .73] | 12.2 |
| | -3 | .35 ± .11 | -.02 [±.12] | 11.8 | .85 [.52, .96] | 34.2 | .80 ± .07 | .02 [±.20] | 7.1 | .34 [-.38, .79] | 23.9 |
| White Matter | +3 | .57 ± .04 | -.01 [±.09] | 4.7 | .57 [-.05, .87] | 14.9 | .98 ± .06 | .00 [±.10] | 2.8 | .76 [.27, .93] | 9.3 |
| | +2 | .52 ± .05 | .00 [±.10] | 4.9 | .59 [-.07, .88] | 17.9 | .97 ± .07 | -.02 [±.12] | 3.1 | .66 [.12, .90] | 12.7 |
| | +1 | .50 ± .07 | -.01 [±.08] | 4.5 | .84 [.47, .96] | 15.0 | .95 ± .04 | -.01 [±.12] | 3.6 | .36 [-.33, .79] | 12.1 |
| | LSE | .48 ± .06 | .00 [±.06] | 2.8 | .91 [.67, .98] | 10.9 | .91 ± .06 | -.03 [±.15] | 5.5 | .40 [-.23, .81] | 16.6 |
| | -1 | .46 ± .06 | .00 [±.07] | 5.5 | .85 [.51, .96] | 15.1 | .92 ± .07 | -.03 [±.20] | 5.3 | .36 [-.30, .79] | 21.1 |
| | -2 | .42 ± .07 | .01 [±.09] | 5.1 | .85 [.50, .96] | 19.1 | .94 ± .08 | -.02 [±.17] | 4.3 | .57 [-.04, .87] | 17.1 |
| | -3 | .42 ± .10 | -.01 [±.12] | 8.8 | .85 [.51, .96] | 26.6 | .93 ± .13 | -.05 [±.28] | 8.5 | .55 [-.04, .86] | 29.6 |
| WM Dorsal | +3 | .64 ± .05 | -.03 [±.12] | 5.7 | .39 [-.19, .80] | 18.6 | 1.01 ± .07 | .02 [±.18] | 5.2 | .42 [-.27, .82] | 17.0 |
| | +2 | .60 ± .05 | .00 [±.14] | 5.8 | .44 [-.29, .83] | 21.8 | .99 ± .08 | .00 [±.15] | 3.8 | .66 [.06, .90] | 14.3 |
| | +1 | .58 ± .07 | -.01 [±.10] | 5.4 | .76 [.28, .94] | 17.1 | .96 ± .06 | .02 [±.14] | 4.4 | .42 [-.24, .82] | 14.7 |
| | LSE | .55 ± .06 | .01 [±.07] | 4.0 | .86 [.55, .96] | 12.3 | .92 ± .06 | .04 [±.18] | 5.6 | .23 [-.34, .72] | 19.8 |
| | -1 | .51 ± .07 | .00 [±.12] | 7.1 | .71 [.16, .92] | 21.8 | .92 ± .14 | .03 [±.27] | 7.6 | .62 [.02, .89] | 28.1 |
| | -2 | .47 ± .08 | .00 [±.08] | 5.2 | .87 [.55, .97] | 16.7 | .91 ± .07 | -.02 [±.18] | 6.2 | .37 [-.33, .80] | 19.4 |
| | -3 | .45 ± .11 | .02 [±.22] | 11.0 | .59 [-.04, .88] | 47.3 | .96 ± .16 | -.01 [±.33] | 9.2 | .58 [-.08, .88] | 32.5 |
| WM Lateral | +3 | .55 ± .04 | .00 [±.08] | 4.2 | .69 [.12, .91] | 13.2 | .96 ± .06 | -.03 [±.13] | 4.5 | .50 [-.06, .84] | 14.1 |
| | +2 | .50 ± .05 | .02 [±.11] | 6.1 | .47 [-.17, .83] | 21.2 | .95 ± .08 | -.04 [±.16] | 4.9 | .57 [.01, .87] | 17.5 |
| | +1 | .48 ± .06 | .01 [±.10] | 5.5 | .73 [.22, .93] | 19.7 | .97 ± .07 | -.04 [±.22] | 5.5 | .28 [-.33, .75] | 22.8 |
| | LSE | .46 ± .07 | .02 [±.10] | 6.4 | .76 [.32, .93] | 20.8 | .93 ± .09 | -.07 [±.21] | 8.3 | .37 [-.17, .78] | 24.7 |
| | -1 | .43 ± .07 | .02 [±.05] | 4.0 | .92 [.69, .98] | 13.5 | .95 ± .08 | -.07 [±.20] | 7.0 | .37 [-.15, .78] | 23.7 |
| | -2 | .42 ± .09 | .02 [±.11] | 7.8 | .80 [.42, .95] | 26.8 | .97 ± .14 | -.04 [±.26] | 7.2 | .63 [.07, .89] | 25.9 |
| | -3 | .43 ± .10 | -.01 [±.16] | 9.7 | .74 [.24, .93] | 34.7 | .93 ± .16 | -.10 [±.41] | 12.7 | .36 [-.21, .78] | 45.8 |
| WM Ventral | +3 | .50 ± .05 | .00 [±.11] | 6.8 | .57 [-.10, .87] | 20.2 | .96 ± .08 | .00 [±.12] | 3.3 | .78 [.33, .94] | 11.6 |
| | +2 | .44 ± .06 | .00 [±.08] | 5.0 | .81 [.39, .95] | 17.0 | .95 ± .08 | -.03 [±.16] | 4.4 | .56 [-.01, .87] | 16.5 |
| | +1 | .43 ± .07 | -.02 [±.07] | 4.7 | .87 [.56, .97] | 16.5 | .93 ± .06 | -.02 [±.10] | 3.1 | .67 [.14, .90] | 10.3 |
| | LSE | .43 ± .06 | -.02 [±.08] | 5.9 | .80 [.41, .95] | 18.7 | .89 ± .06 | -.05 [±.14] | 5.5 | .45 [-.10, .82] | 16.8 |
| | -1 | .43 ± .06 | -.02 [±.09] | 6.6 | .70 [.22, .92] | 21.1 | .90 ± .05 | -.03 [±.17] | 5.8 | .14 [-.44, .67] | 19.0 |
| | -2 | .39 ± .07 | -.01 [±.12] | 7.4 | .70 [.16, .92] | 28.2 | .93 ± .09 | -.01 [±.12] | 3.7 | .81 [.40, .95] | 11.8 |
| | -3 | .36 ± .11 | -.04 [±.13] | 15.5 | .78 [.32, .94] | 40.5 | .88 ± .13 | -.02 [±.26] | 8.0 | .60 [-.02, .89] | 27.7 |

* Indicates significant difference between scan and rescan (p < 0.05).

*Notes*: The individual axial slice stacks were aligned at the LSE landmark, defined as the slice with the largest gray matter CSA ($GM_{max,mw}$), and were adjusted for the length of the conus medullaris. The landmarks were determined in the first scan. A positive segment indicates a rostral direction from the LSE landmark.

*Abbreviations*: CI, confidence interval; CV, scan-rescan coefficient of variation; $\bar{d}$, mean scan-rescan difference; ICC, scan-rescan intraclass correlation coefficient; LSE, lumbosacral enlargement; MDC, minimal detectable change; SD, standard deviation; WM, white matter.

neurological levels as long as their distribution within each segment remains constant across subjects. In addition, while the adjustment method was demonstrated on ME-GRE images, it can be applied to other MRI sequences as well.

**Table 5. Scan-rescan reliability of axial and radial diffusivity values (n = 10 healthy volunteers).**

| | | Axial Diffusivity ($10^{-3}$ mm²/s) | | | | | Radial Diffusivity ($10^{-3}$ mm²/s) | | | | |
|---|---|---|---|---|---|---|---|---|---|---|---|
| | Segment | mean ± SD | $\bar{d}$ [± 1.96 SD] | CV (%) | ICC [95% CI] | MDC (%) | mean ± SD | $\bar{d}$ [± 1.96 SD] | CV (%) | ICC [95% CI] | MDC (%) |
| Gray Matter | +3 | 1.40 ± .16 | .03 [±.23] | 4.9 | .76 [.30, .93] | 15.9 | .65 ± .03 | .02 [±.08] | 4.2 | .36 [-.26, .79] | 12.7 |
| | +2 | 1.28 ± .15 | .03 [±.17] | 3.7 | .85 [.53, .96] | 13.3 | .68 ± .03 | .00 [±.16] | 5.6 | -.20 [-.84, .50] | 23.0 |
| | +1 | 1.19 ± .07 | .00 [±.13] | 3.3 | .67 [.08, .91] | 10.5 | .69 ± .05 | .02 [±.10] | 3.6 | .58 [-.01, .87] | 14.2 |
| | LSE | 1.11 ± .09 | .00 [±.19] | 5.1 | .57 [-.10, .87] | 15.9 | .65 ± .05 | .00 [±.11] | 5.0 | .48 [-.23, .84] | 15.8 |
| | -1 | 1.10 ± .09 | -.01 [±.22] | 5.6 | .43 [-.30, .82] | 19.1 | .66 ± .03 | .01 [±.14] | 6.6 | -.12 [-.79, .55] | 21.1 |
| | -2 | 1.10 ± .10 | -.01 [±.16] | 4.1 | .75 [.26, .93] | 13.4 | .67 ± .04 | -.01 [±.12] | 4.8 | .22 [-.51, .74] | 16.8 |
| | -3 | 1.12 ± .15 | -.01 [±.34] | 8.5 | .55 [-.13, .87] | 28.4 | .64 ± .08 | .03 [±.16] | 7.0 | .59 [.02, .88] | 25.5 |
| White Matter | +3 | 1.70 ± .17 | -.02 [±.15] | 2.2 | .90 [.66, .97] | 8.6 | .62 ± .03 | .00 [±.13] | 5.8 | .00 [-.71, .63] | 19.9 |
| | +2 | 1.61 ± .19 | -.02 [±.12] | 2.3 | .94 [.80, .99] | 7.2 | .64 ± .04 | -.02 [±.16] | 5.7 | -.09 [-.70, .55] | 24.0 |
| | +1 | 1.54 ± .13 | -.02 [±.14] | 2.7 | .85 [.54, .96] | 8.9 | .66 ± .06 | -.01 [±.13] | 5.7 | .53 [-.14, .86] | 19.4 |
| | LSE | 1.45 ± .14 | -.02 [±.22] | 4.7 | .73 [.24, .93] | 14.2 | .65 ± .06 | -.03 [±.13] | 6.4 | .48 [-.11, .84] | 20.8 |
| | -1 | 1.43 ± .17 | -.04 [±.28] | 5.4 | .71 [.20, .92] | 19.1 | .67 ± .05 | -.02 [±.17] | 5.7 | .19 [-.47, .71] | 24.9 |
| | -2 | 1.40 ± .15 | -.02 [±.25] | 4.7 | .72 [.20, .92] | 16.6 | .71 ± .08 | -.03 [±.14] | 5.0 | .64 [.10, .89] | 19.9 |
| | -3 | 1.37 ± .22 | -.08 [±.38] | 7.7 | .65 [.13, .90] | 28.0 | .70 ± .12 | -.03 [±.25] | 10.2 | .59 [-.01, .88] | 34.4 |
| WM Dorsal | +3 | 1.87 ± .17 | -.02 [±.23] | 3.6 | .81 [.40, .95] | 11.5 | .58 ± .05 | .04 [±.21] | 10.5 | .01 [-.58, .61] | 35.5 |
| | +2 | 1.78 ± .19 | .00 [±.20] | 3.2 | .88 [.58, .97] | 10.5 | .59 ± .06 | .00 [±.20] | 8.5 | .25 [-.50, .75] | 31.6 |
| | +1 | 1.67 ± .14 | .03 [±.17] | 2.9 | .83 [.47, .95] | 10.2 | .60 ± .07 | .01 [±.17] | 7.7 | .50 [-.18, .85] | 27.4 |
| | LSE | 1.57 ± .13 | .09 [±.24] | 5.1 | .56 [-.01, .86] | 17.0 | .60 ± .07 | .02 [±.16] | 7.8 | .50 [-.15, .85] | 26.1 |
| | -1 | 1.50 ± .23 | .04 [±.36] | 6.6 | .75 [.27, .93] | 22.8 | .63 ± .12 | .02 [±.25] | 11.4 | .56 [-.09, .87] | 37.5 |
| | -2 | 1.42 ± .16 | -.02 [±.26] | 6.1 | .71 [.18, .92] | 17.5 | .66 ± .06 | -.02 [±.16] | 6.2 | .42 [-.25, .82] | 23.4 |
| | -3 | 1.46 ± .21 | .02 [±.38] | 6.6 | .67 [.10, .91] | 25.0 | .71 ± .16 | -.03 [±.36] | 15.8 | .54 [-.12, .87] | 48.3 |
| WM Lateral | +3 | 1.62 ± .15 | -.04 [±.21] | 3.3 | .75 [.30, .93] | 13.1 | .62 ± .04 | -.03 [±.13] | 6.9 | .24 [-.34, .72] | 21.5 |
| | +2 | 1.55 ± .18 | -.03 [±.16] | 2.7 | .90 [.68, .97] | 9.9 | .65 ± .05 | -.05 [±.19] | 8.3 | -.04 [-.51, .53] | 29.7 |
| | +1 | 1.52 ± .14 | -.05 [±.22] | 4.1 | .72 [.24, .92] | 14.8 | .69 ± .08 | -.04 [±.23] | 7.9 | .26 [-.37, .74] | 33.5 |
| | LSE | 1.44 ± .16 | -.07 [±.28] | 6.5 | .64 [.11, .89] | 20.3 | .68 ± .08 | .-07 [±.20] | 10.8 | .33 [-.18, .76] | 32.6 |
| | -1 | 1.42 ± .19 | -.07 [±.30] | 6.3 | .68 [.18, .91] | 21.8 | .71 ± .06 | -.07 [±.17]* | 8.0 | .26 [-.20, .71] | 27.2 |
| | -2 | 1.43 ± .21 | -.02 [±.34] | 6.2 | .74 [.23, .93] | 22.2 | .74 ± .13 | -.05 [±.24] | 9.0 | .62 [.08, .89] | 32.3 |
| | -3 | 1.40 ± .26 | -.16 [±.55] | 11.2 | .50 [-.05, .84] | 41.7 | .70 ± .14 | -.07 [±.38] | 14.7 | .36 [-.26, .78] | 54.1 |
| WM Ventral | +3 | 1.56 ± .20 | .01 [±.29] | 4.3 | .77 [.30, .94] | 17.4 | .66 ± .04 | .00 [±.10] | 4.8 | .49 [-.22, .85] | 14.6 |
| | +2 | 1.47 ± .19 | -.05 [±.20] | 4.0 | .85 [.52, .96] | 14.1 | .69 ± .04 | -.02 [±.16] | 6.0 | .09 [-.58, .66] | 22.5 |
| | +1 | 1.41 ± .14 | -.05 [±.14] | 3.2 | .84 [.42, .96] | 11.2 | .69 ± .06 | .00 [±.10] | 4.2 | .76 [.26, .93] | 13.3 |
| | LSE | 1.35 ± .15 | -.08 [±.22]* | 6.1 | .67 [.11, .91] | 18.9 | .66 ± .05 | -.03 [±.12] | 5.7 | .44 [-.14, .82] | 18.5 |
| | -1 | 1.37 ± .14 | -.07 [±.24] | 5.9 | .63 [.09, .89] | 18.7 | .67 ± .03 | -.02 [±.15] | 6.1 | -.09 [-.71, .56] | 22.0 |
| | -2 | 1.36 ± .17 | -.02 [±.22] | 4.8 | .81 [.42, .95] | 15.3 | .72 ± .07 | .00 [±.11] | 4.6 | .78 [.31, .94] | 14.0 |
| | -3 | 1.25 ± .22 | -.08 [±.27] | 6.2 | .78 [.33, .94] | 23.2 | .70 ± .12 | .01 [±.27] | 11.5 | .56 [-.11, .87] | 36.2 |

\* Indicates significant difference between scan and rescan (p < 0.05).

*Notes*: The individual axial slice stacks were aligned at the LSE landmark, defined as the slice with the largest gray matter CSA ($GM_{max,mw}$), and were adjusted for the length of the conus medullaris. The landmarks were determined in the first scan. A positive segment indicates a rostral direction from the LSE landmark.

*Abbreviations*: CI, confidence interval; CV, scan-rescan coefficient of variation; $\bar{d}$, mean scan-rescan difference; ICC, scan-rescan intraclass correlation coefficient; LSE, lumbosacral enlargement; MDC, minimal detectable change; SD, standard deviation; WM, white matter.

### 4.3 Excellent intra-rater reliability of cross-sectional area measurements at the lumbosacral enlargement

We demonstrated excellent intra-rater reliability for CSA measurements, when using the previously optimized ME-GRE imaging protocol [17], with intra-rater CV values below 4%, 5%, and 6% for SC, GM, and WM CSA, respectively, at and within a 15 mm radius of the LSE landmark. Notably, the inter-rater CV values for CSA measurements were approximately twice as high as the corresponding intra-rater CV values. Consequently, we strongly advocate for the segmentation of all images by the same rater.

We also highlight the importance of anatomical location for the reliability of CSA measurements. More caudal slices within the CM had lower intra- and inter-rater reliability, in line with a previous report [18]. This observation is likely attributed to the conus medullaris becoming thinner; for example, a minor difference between two segmentations has a larger impact when the area being segmented is smaller. Despite the increasing CV values toward the tip of the spinal cord, the intra- and inter-rater ICC values exhibited relatively consistent trends along the CM. This is because higher CV values are offset by higher inter-subject variability at more caudal locations (Table 1).

While intra-rater CV values for SC, GM, and WM CSA measurements were, on average, less than 1 percentage point higher in patients with spinal cord injury compared to healthy controls, the inter-rater CV values were, on average, between 2 and 5 percentage points higher, with the largest differences occurring caudal to the LSE landmark. This finding suggests that, for patients, the lower quality of the ME-GRE images increases the inter-rater, but not the intra-rater variability of SC and GM segmentations. The lower quality for patients may be attributed to a higher level of involuntary motion (4 of 5 patients reported spasticity in the lower limbs). While 4 of 5 patients had spinal instrumentation, they were not in close proximity to our field of view (FOV) and are therefore unlikely to affect the image quality.

The higher values for SC CSA, in comparison to GM and WM CSA, reflect the experience of the raters that SC segmentation is an "easier" task than GM segmentation. This is due to the fact that the contrast between the GM and WM is typically lower than between the WM and cerebrospinal fluid on ME-GRE images, and GM exhibits more ambiguity owing to its irregular shape [17]. The reliability of CSA measurements depends on the confidence with which manual segmentation is performed, referred to as "segmentability". SC and GM segmentability is influenced by various factors, including the rater's experience, the subject being investigated, the imaging hardware, pulse sequence, and sequence parameters [17]. Consequently, the comparability of reliability values across studies is limited by variations in these factors. In comparison to values obtained on a 3T Philips MRI scanner [18], our study showed lower intra-rater (1.6% vs. 3.5%) but higher inter-rater CV for SC CSA (4.7% vs. 3.0%) at the LSE landmark within healthy volunteers. The CV values for GM CSA were found to be lower in our study for both intra-rater (4.0% vs. 7.7%) and inter-rater (6.1% vs. 9.9%) analyses.

### 4.4 Scan-rescan reliability depends on the anatomical location within the lumbosacral cord

Scan-rescan reliability was generally lower compared to the corresponding intra-rater reliability. This is expected, as in addition to the variabilities captured by intra-rater reliability, scan-rescan reliability also encompasses variabilities arising from subject positioning, the position of the imaged organ, FOV positioning, and potential imaging hardware instabilities (e.g., scanner drift). The time interval between scan and rescan is another important factor; longer intervals are associated with lower reliability. In our study, we deliberately selected a relatively long time interval of 6 to 15 weeks to mimic time frames commonly encountered in longitudinal

clinical studies. Scan-rescan reliability was generally higher compared to the corresponding inter-reliability, which further emphasizes the importance of employing the same rater for segmentation.

The scan-rescan reliability of DTI metrics was found to be higher (i) in slices with larger CSA, particularly at and around the LSE landmark, as opposed to more caudal slices, and (ii) within larger regions (such as WM), compared to smaller ones (such as dorsal and ventral WM). This is because averaging DTI metrics within a larger region reduces the impact of individual outliers, small anatomical variations, and partial volume effects. Scan-rescan reliability can be further improved by averaging DTI metrics across several adjacent slices or segments, although this comes at the cost of decreased spatial specificity. For example, when averaging across three adjacent slices around the LSE landmark, as in [23,25], we observed a reduction in scan-rescan CV from 3.4% to 2.2% for WM CSA and from 4.2% to 2.7% for FA within the WM. When adjusting for the CM length, a small improvement in the scan-rescan reliability for DTI metrics was observed, which is probably due to the larger thickness of the segments (1–2 slices) compared to a single slice.

The DTI metrics exhibited lower scan-rescan reliability in comparison to CSA measurements. This is because diffusion MRI is inherently noisier than structural MRI, and, unlike CSA measurements, the average DTI metrics within a ROI are not solely determined by the size of the ROI, but also by its location. For example, if two SC segmentations do not fully overlap but have the same area, they would yield the same SC CSA, yet different average DTI metrics within the SC. Nevertheless, we demonstrated a very high overlap between segmentation masks created by the same rater, as measured by the Dice coefficient, with values above 91% for SC, 90% for GM, and 76% for WM segmentation.

Despite differences in the cohort, MRI scanner, sequence, and processing pipeline, our scan-rescan CV for SC CSA at the LSE landmark is similar to that reported previously in healthy volunteers scanned on a 3T Philips MRI scanner (vs. Yiannakas et al., 2014 [23]; 1.8% vs. 2.0%). However, our CV for GM CSA was notably lower (2.9% vs. 7.8%). For DTI metrics extracted within the WM at the LSE landmark, our CV values were in line with previously reported values for FA (vs. Yiannakas et al., 2016 [25]: 2.8% vs. 6.0%), MD (5.5% vs. 5.0%), AD (4.7% vs. 5.4%), and RD (6.4% vs. 8.3%). The provided scan-rescan reliability values (Tables 3–5) serve as valuable guides for power and sample size calculations in future longitudinal studies. For example, we can calculate, using the equations in [41], that detecting a 3% change over time in GM CSA at the LSE with a power of 80% and significance of 5% (one-sided) would require 12 subjects.

## 4.5 Limitations

In contrast to CSA values, we did not separately investigate intra- and inter-rater reliability for DTI metrics. This is because while obtaining CSA values involves a single manual step (SC and GM segmentation in the ME-GRE images), the processing pipeline for DTI requires several manual interventions (selection of SC midpoint, SC segmentation in both the EPI and ME-GRE images), making it challenging to isolate individual effects. However, we argue that the reported scan-rescan reliability values encompass the combined effect of all these sources of variability and are therefore most relevant for planning future studies. Additionally, DTI metrics could not be extracted from the caudal half of the CM, as this region is currently not covered by the PAM50 atlas. This is a recognized issue within the community and is likely to be addressed in future developments. A limitation of the study is the relatively small sample size; however, this sample size is typical in method development and scan-rescan studies. Moreover, the findings may not be generalizable to patient populations with pathologies (e.g. lesions) within the lumbosacral

cord. Another limitation is the considerable length of the MRI examination; however, the acquisition time of the ME-GRE images can be reduced for clinical applications [17].

## 5. Conclusions

We provide recommendations for improved inter-subject alignment within the lumbosacral cord to facilitate group-level analyses of MRI metrics. Specifically, we propose using the slice with the largest gray matter cross-sectional area as a reliable image-based neuroanatomical landmark, along with an adjustment method for the length of the conus medullaris. We emphasize the importance of anatomical location for intra-rater, inter-rater, and scan-rescan reliability, which were the highest at the lumbosacral enlargement and decreased toward the tip of the spinal cord. The provided scan-rescan reliability values serve as valuable guides for power and sample size calculations in future longitudinal studies.

## Supporting information

**S1 Fig. Bland-Altman plots illustrating the scan-rescan differences of cross-sectional area measurements and diffusion tensor imaging metrics within the white matter (WM), at the lumbosacral enlargement (segment LSE, in blue) and the middle of the conus medullaris (segment LSE-3, in red) (n = 10 healthy volunteers).** The dotted lines represent the bias, i.e., the mean of the scan-rescan differences, while the dashed lines represent the 95% limits of agreement.
(TIF)

**S1 Table. Demographic and clinical information of patients with spinal cord injury.**
(DOCX)

**S2 Table. Slice-wise cross-sectional area values when aligning the individual slice stacks at different lumbosacral enlargement landmarks.**
(DOCX)

**S3 Table. Slice-wise intra- and inter-rater reliability of cross-sectional area measurements (n = 10; 5 healthy controls and 5 patients with spinal cord injury).**
(DOCX)

**S4 Table. Slice-wise scan-rescan reliability of cross-sectional area measurements (n = 10 healthy volunteers).**
(DOCX)

**S5 Table. Slice-wise scan-rescan reliability of fractional anisotropy and mean diffusivity values (n = 10 healthy volunteers).**
(DOCX)

**S6 Table. Slice-wise scan-rescan reliability of axial and radial diffusivity values (n = 10 healthy volunteers).**
(DOCX)

**S7 Table. Scan-rescan reliability of cross-sectional area measurements (n = 10 healthy volunteers).**
(DOCX)

**S8 Table. Scan-rescan reliability of fractional anisotropy and mean diffusivity values (n = 10 healthy volunteers).**
(DOCX)

**S9 Table. Scan-rescan reliability of axial and radial diffusivity values (n = 10 healthy volunteers).**
(DOCX)

## Acknowledgments

We thank all the volunteers who participated in this study. We also thank Veronika Birkhäuser and Oliver Gross, as well as the entire clinical team of the Department of Neuro-Urology, Balgrist University Hospital, University of Zürich for recruiting the patients and screening the healthy volunteers. Furthermore, we thank Collene Anderson and Adrian Cathomen for pseudo-randomizing the images for segmentation. We also thank all members of the TASCI Study Group for their support. Imaging was performed with support of the Swiss Center for Musculoskeletal Imaging, SCMI, Balgrist Campus AG, Zürich.

## Author Contributions

**Conceptualization:** Silvan Büeler, Patrick Freund, Thomas M. Kessler, Martina D. Liechti, Gergely David.

**Formal analysis:** Silvan Büeler, Gergely David.

**Investigation:** Silvan Büeler, Martina D. Liechti, Gergely David.

**Project administration:** Silvan Büeler.

**Software:** Gergely David.

**Supervision:** Patrick Freund, Thomas M. Kessler, Martina D. Liechti, Gergely David.

**Visualization:** Silvan Büeler, Gergely David.

**Writing – original draft:** Silvan Büeler, Gergely David.

**Writing – review & editing:** Patrick Freund, Thomas M. Kessler, Martina D. Liechti.

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
