## [Decision Letter · Decision Letter 0]

30 Jan 2024

PONE-D-23-37329Improved inter-subject alignment of the lumbosacral cord for group-level in vivo gray and white matter assessments: A scan-rescan MRI study at 3TPLOS ONE

Dear Dr. David,

Thank you for submitting your manuscript to PLOS ONE. After careful consideration, we feel that it has merit but does not fully meet PLOS ONE’s publication criteria as it currently stands. Therefore, we invite you to submit a revised version of the manuscript that addresses the points raised during the review process.

**the article need some English editing**

**the sample size are really too small and not representative **

**Please clarify the blinding process during the assessment**

**the author need to submit the row data**

**the figures are really bad ... they need to be enhance (specially figure 1,5 and 6)**

We look forward to receiving your revised manuscript.

Kind regards,

Ramada Rateb Khasawneh

Academic Editor

PLOS ONE

Journal Requirements:

This work is financially supported by the Swiss National Science Foundation (SNSF) (33IC30_179644). PF is funded by a SNSF Eccellenza Professorial Fellowship grant (PCEFP3_181362/1).

Additional Editor Comments:

the article need some English editing

the sample size are really too small and not representative

Please clarify the blinding process during the assessment

the author need to submit the row data

the figures are really bad ... they need to be enhance (specially figure 1,5 and 6)

Reviewers' comments:

Reviewer's Responses to Questions

**Comments to the Author**

1. Is the manuscript technically sound, and do the data support the conclusions?

Reviewer #1: Yes

Reviewer #2: Yes

2. Has the statistical analysis been performed appropriately and rigorously? 

Reviewer #1: Yes

Reviewer #2: Yes

3. Have the authors made all data underlying the findings in their manuscript fully available?

Reviewer #1: Yes

Reviewer #2: No

4. Is the manuscript presented in an intelligible fashion and written in standard English?

Reviewer #1: Yes

Reviewer #2: Yes

5. Review Comments to the Author

Reviewer #1: Within this manuscript, the authors firstly aim to identify measures of improving the inter-subject alignment for MRI of the lumbosacral spinal cord by investigating a set of 10 healthy control persons and 5 patients with spinal injuries. Moreover, they asses the scan-rescan reliability of the spinal cord imaging metrics by performing a re-scan of healthy controls.

Overall, I believe that this is a well-written manuscript which provides new strategies to improve MR imaging in the lumbosacral cord, which may ultimately contribute to and facilitate design in upcoming studies.

Strengths: The authors use multiple independent investigators and perform a rigorous statistical evaluation pertaining to the individual aims pre-specified in the background section.

Weaknesses: Both the healthy and the patient sample sizes are very small, thus overall limiting the deductions made. Moreover, the patient group is highly heterogeneous, therefore, the conclusions derived from the patients with spinal disease are not clear. These limitations should be made more clear throughout the manuscript.

Questions:

1. For the segmentation of total cord, gray, and white matter measures, it seems as if the raters were trained on three training subjects only. Further down in the manuscript, the three raters re described as "experienced". I suggest clarifying how much spinal cord segmentation experience the three raters actually do have, in order to increase credibility of the presented data.

2. Please clarify the blinding process during the assessment: How was adequate blinding of raters to participant status ensured? Were measures taken to blind signs of spinal injury/disease?

3. Within the five patients with spinal diseases included in this study, there is considerable variability between the nature of the injury/disease (ranging from spinal ischemia to inflammation to dislocation fractures possibly leading to spinal cord compression), and moreover, the location of said spinal injury. One might argue that this high variability of pathologies may influence the segmentation process and introduce potential confounding sources. How were the patients chosen? Would an analysis using only healthy people have sufficed or maybe eben presented a more solid sample to reach this study's aims?

4. The considerable length of the MRI scan per person should be listed as a limitation, seeing as the authors correctly state that lying still in the MRI may be challenging, especially for patients.

5. The sample size in this study is very small, which should be addressed in the limitations section, as should the heterogeneity of the patient sample.

Minor edits:

1. Introduction: "Cross-sectional area (CSA) measurements of SC, GM, and WM, derived from multi-echo gradient-echo sequences, has been utilized as indirect measures of atrophy in the cervical cord and lumbosacral enlargement (LSE) (6–12)." Correct "has" to "have".

Reviewer #2: Firstly, I would like to express my appreciation for the opportunity to review your manuscript entitled "Improved inter-subject alignment of the lumbosacral cord for group-level in vivo gray and white matter assessments: A scan-rescan MRI study at 3T." This study is a commendable effort in advancing our understanding of MRI assessments in the lumbosacral spinal cord, addressing significant technical challenges in the field.

The manuscript presents a detailed and methodologically sound approach, offering valuable insights into the reliability and accuracy of MRI metrics. The innovative methods and analyses employed in your study are particularly noteworthy. However, to further enhance the impact and clarity of the manuscript, there are several areas that would benefit from additional attention and refinement. These include expanding on the practical implications of your findings, providing more detailed methodological descriptions, and a deeper engagement with existing literature.

Enclosed in the attached PDF is a detailed review that elaborates on these points, along with suggestions for potential improvements and considerations for future research.

I am looking forward to witnessing the continued development of your work and its contributions to the field of MRI research.

6. PLOS authors have the option to publish the peer review history of their article (what does this mean?). If published, this will include your full peer review and any attached files.

Reviewer #1: No

Reviewer #2: **Yes: **R. Heller

---

## [Author Response · Author response to Decision Letter 0]

21 Feb 2024

Point-by-point response to reviewers

We would like to thank the reviewers for taking their time to review our manuscript and for providing constructive comments. We have addressed them point-by-point as follows:

Reviewer 1

Within this manuscript, the authors firstly aim to identify measures of improving the inter-subject alignment for MRI of the lumbosacral spinal cord by investigating a set of 10 healthy control persons and 5 patients with spinal injuries. Moreover, they asses the scan-rescan reliability of the spinal cord imaging metrics by performing a re-scan of healthy controls.

Overall, I believe that this is a well-written manuscript which provides new strategies to improve MR imaging in the lumbosacral cord, which may ultimately contribute to and facilitate design in upcoming studies.

Strengths: The authors use multiple independent investigators and perform a rigorous statistical evaluation pertaining to the individual aims pre-specified in the background section.

Weaknesses: Both the healthy and the patient sample sizes are very small, thus overall limiting the deductions made. Moreover, the patient group is highly heterogeneous, therefore, the conclusions derived from the patients with spinal disease are not clear. These limitations should be made more clear throughout the manuscript.

Questions:

1. For the segmentation of total cord, gray, and white matter measures, it seems as if the raters were trained on three training subjects only. Further down in the manuscript, the three raters are described as "experienced". I suggest clarifying how much spinal cord segmentation experience the three raters actually do have, in order to increase credibility of the presented data.

We thank the reviewer for highlighting this potential source of misunderstanding. We have now added a sentence to clarify the experience of the raters:

Section 2.3: "The raters had extensive experience in manually segmenting the lumbosacral cord, having individually segmented over 100 images prior to this study. To establish consensus guidelines for segmentation, the three raters initially segmented a separate training set comprising three healthy volunteers, which was not included in the main analysis."

2. Please clarify the blinding process during the assessment: How was adequate blinding of raters to participant status ensured? Were measures taken to blind signs of spinal injury/disease?

We thank the reviewer for this relevant question. We have now clarified the blinding process as follows:

Section 2.7: "Raters were provided with the pseudonymized set of images (ME-GRE of the lumbosacral cord). The raters were not provided with images covering the injury site, where the participant status would have been visible. Since the patients did not exhibit any injury-related radiological abnormalities or implant-related artifacts in the lumbosacral cord, raters were unable to discern whether the participant was healthy or a patient."

3. Within the five patients with spinal diseases included in this study, there is considerable variability between the nature of the injury/disease (ranging from spinal ischemia to inflammation to dislocation fractures possibly leading to spinal cord compression), and moreover, the location of said spinal injury. One might argue that this high variability of pathologies may influence the segmentation process and introduce potential confounding sources. How were the patients chosen? Would an analysis using only healthy people have sufficed or maybe even presented a more solid sample to reach this study's aims?

Segmentation reliability values obtained in healthy participants might not generalize to patients because they frequently experience higher levels of involuntary motion (e.g., due to spasticity), which could impact image quality and consequently the image segmentability. Since the presented sequences are often used in clinical research studies, we chose to include an additional (albeit small) group of patients with spinal cord injury (SCI), which represents a more typical group encountered in clinical studies. Patients with SCI are a challenging imaging cohort, often exhibiting involuntary movements.

While we acknowledge that a pathology such as a lesion affecting the lumbosacral cord would significantly impact the segmentation process, it's important to clarify that our primary aim was not to assess this variability. Instead, our focus was on measuring segmentation reliability in the radiologically normal appearing lumbosacral cord of patients. Therefore, patients were selected based on the absence of injury-related radiological structural abnormalities in the lumbosacral cord. We clarified this point in the introduction:

Section 1: "Here, our focus was on measuring the reliability of MRI metrics in a patient cohort where the lumbosacral cord is not directly affected. For this reason, we included patients with a spinal cord injury (SCI) in the cervical or thoracic cord, who represent a challenging imaging cohort as they often experience higher levels of involuntary motion (e.g., due to spasticity)."

Consequently, we argue that the heterogeneity of the patients does not influence the segmentation process of the lumbosacral cord, as the injuries were located well above in the cervical or thoracic cord. In fact, it was not possible to distinguish patients and controls based on the images. It also means that our findings cannot generalize to patient populations with pathologies within the lumbosacral cord. We added a sentence in the Limitations section clarifying this point:

Section 4.5: "Moreover, the findings may not be generalizable to patient populations with pathologies (e.g. lesions) within the lumbosacral cord."

For the scan-rescan analysis, it was essential to include only healthy participants, as stated in section 2.8: "The reliability analysis across both imaging sessions (scan and rescan) included the 10 healthy volunteers. Patients were not included, as their scan-rescan data might be affected by disease-related longitudinal changes."

4. The considerable length of the MRI scan per person should be listed as a limitation, seeing as the authors correctly state that lying still in the MRI may be challenging, especially for patients.

We agree with the reviewer and have addressed this concern in the Limitation section. Specifically, we reference our previous work where we demonstrated that the acquisition time could be reduced to 10:38 min for clinical applications, while still maintaining good contrast for reliable gray matter and spinal cord segmentation.

Section 4.5: "Another limitation is the considerable length of the MRI examination; however, the acquisition time of the ME-GRE images can be reduced for clinical applications (17)."

5. The sample size in this study is very small, which should be addressed in the limitations section, as should the heterogeneity of the patient sample.

We have added a sentence to the Limitations section:

Section 4.5: " A limitation of the study is the relatively small sample size; however, this sample size is typical in method development and scan-rescan studies." 

Regarding the heterogeneity, similar to question #3, we argue that this is per se not a limitation of this specific study because the lumbosacral cord is not directly affected by the injury. Our motivation in including patients was that they might behave differently in the scanner (e.g., motion artifacts), which could impact data quality and subsequent image segmentability. In other words, we aimed to include a (small) cohort that is more representative to clinical populations than healthy controls. In all patients, the injury itself was clearly above the region of interest and was not the focus of the study.

Minor edits:

1. Introduction: "Cross-sectional area (CSA) measurements of SC, GM, and WM, derived from multi-echo gradient-echo sequences, has been utilized as indirect measures of atrophy in the cervical cord and lumbosacral enlargement (LSE) (6–12)." Correct "has" to "have".

Thank you for pointing out this mistake. We corrected "has" to "have".

 

Reviewer 2

Firstly, I would like to express my appreciation for the opportunity to review your manuscript entitled "Improved inter-subject alignment of the lumbosacral cord for group-level in vivo gray and white matter assessments: A scan-rescan MRI study at 3T." This study is a commendable effort in advancing our understanding of MRI assessments in the lumbosacral spinal cord, addressing significant technical challenges in the field.

The manuscript presents a detailed and methodologically sound approach, offering valuable insights into the reliability and accuracy of MRI metrics. The innovative methods and analyses employed in your study are particularly noteworthy. However, to further enhance the impact and clarity of the manuscript, there are several areas that would benefit from additional attention and refinement. These include expanding on the practical implications of your findings, providing more detailed methodological descriptions, and a deeper engagement with existing literature.

Enclosed in the attached PDF is a detailed review that elaborates on these points, along with suggestions for potential improvements and considerations for future research.

I am looking forward to witnessing the continued development of your work and its contributions to the field of MRI research.

Major Comments:

1. Innovative Approach: The strength of the study lies in its innovative approach to a complex problem in MRI analysis. However, a more detailed discussion on how this approach can be generalized or applied in other contexts would additionally strengthen the manuscript.

While the reported inter-rater, inter-rater, and scan-rescan reliability values are specific to the lumbosacral cord and the presented sequences (ME-GRE sequence and DTI), the proposed method for adjusting for the length of the conus medullaris can be applied to any MRI sequence. Therefore, we have added the following sentence:

Section 4.2: "In addition, while the adjustment method was demonstrated on ME-GRE images, it can be applied to other MRI sequences as well.

2. Sample Diversity: The inclusion of both healthy individuals and patients with spinal cord injury adds depth to the study. However, the manuscript could benefit from a further discussion on the potential biases or limitations this sample diversity might introduce.

We appreciate the reviewer's suggestion. We have addressed this point in detail in our response to question #3 from Reviewer 1. As detailed there, we do not consider the sample diversity to be a limiting factor, given that the primary focus of the study was not the injury itself. Instead, patients were included due to potential variations in behavior during scanning (e.g., motion artifacts), which could impact data quality and subsequent image segmentability.

In discussing the differences between patients and healthy participants, we have highlighted in section 4.3 that a higher coefficient of variation (CV) was observed in patients compared to healthy participants for inter-rater analysis, while no significant differences were found for intra-rater analysis.

3. Reliability Measures: The study presents compelling reliability measures, hence outlining concepts for the requirements in a multi-centre design (e.g. sample size) would enhance the manuscript’s rigor.

We agree with the reviewer and added a section in the discussion on sample size calculation.

Section 4.4: "The provided scan-rescan reliability values (Tables 3-5) serve as valuable guides for power and sample size calculations in future longitudinal studies. For example, we can calculate, using the equations in (41), that detecting a 3% change over time in GM CSA at the LSE with a power of 80% and significance of 5% (one-sided) would require 12 subjects."

4. Technical Limitations and Challenges: Delving deeper into the technical challenges and limitations encountered during the study, especially concerning MRI techniques and measurements, would provide valuable insights for future research.

The technical challenges of the applied sequences were extensively addressed in our previous publication (Büeler et al., 2022; Reference #17). In the Limitation section of the current paper, we discuss additional challenges such as the length of the scan.

5. Impact of Participant Characteristics: Given a limited sample size, the study should consider how further individual differences in participants, such as age, gender, or specific severity of spinal cord injury, might influence the MRI metrics.

The MRI metrics indeed show variation with age, gender, and other characteristics. However, it's important to note that the focus of this study was not on these metrics themselves, but rather on their scan-rescan variability, which is not expected to be that significant.

We also note that accounting for these variations is of great interest to identify pathological values, for example. We shortly acknowledge this in the Discussion:

Section 4.2: "We note that another line of research is concerned with reducing inter-subject variability of MRI metrics that arise from biological variability, using regression models incorporating demographic information, spine, and SC metrics (18,39); however, this was not the focus of our investigation."

6. Future Research Directions: Finally, the study should outline further potential directions for future research, especially in exploring other methods of alignment and landmark identification in MRI studies, and their clinical applicability.

We elaborate on this point in the Discussion section:

Section 4.1: "Notably, a previous study based on post-mortem MRI utilized distinct morphological features of the ventral GM horns to characterize the lumbosacral cord across species (36). Another study revealed a strong link between these morphological features and the motor neuron pools located within the ventral GM horns (37). We anticipate that, for in vivo studies, the shape of GM would serve as an even better neuroanatomical landmark than its size. In a recent study, a distinct approach was used to directly identify neurological levels through nerve root tracing (38). However, this approach necessitates acquiring an additional image optimized for nerve segmentation, thereby increasing the scan time, and might not work reliably in all subjects."

Minor Comments

With appreciation for the study’s strengths, these points are provided with the intent of further enriching the publication. From this reviewer’s perspective, they are not mandatory for publication but are proposed as enhancements that could potentially add depth and clarity to your already substantial work.

1. Detailed Reporting of Technical Parameters and code of analysis: While the study’s methodology is robust, and data can be accessed via github, providing more detailed descriptions of the technical parameters used in MRI assessments (e.g., specific settings or protocols including the analysis code) would significantly enhance the replicability and comprehension of the methods.

We appreciate the reviewer's suggestion. We have uploaded the MRI exam cards, which contain all MRI sequence parameters, to GitHub. The link can be accessed in Section 2.3. This will allow researchers to reproduce our sequences on their respective scanners.

2. Consideration of Technical Variabilities: Discussing the potential technical variabilities that might impact the MRI measurements, such as machine calibration or operator expertise, would provide a more comprehensive view of the challenges in achieving reliable MRI metrics.

The Discussion section highlights some of the technical variabilities that impact MRI metrics:

Section 4.4: "scan-rescan reliability also encompasses variabilities arising from subject positioning, the position of the imaged organ, FOV positioning, and potential imaging hardware instabilities (e.g., scanner drift)."

3. Suggestions for Future Research: Further offering specific suggestions for future research directions, particularly in areas that were not fully explored in the current study, would be valuable for advancing the field.

In the present paper, we propose a method for adjusting for the length of conus medullaris based on two landmarks. While technically not feasible currently, an even better approach would be to identify neurological levels directly on the axial ME-GRE images. This is now proposed as future research:

Section 4.1: "Future research should focus on identifying neurological levels directly from the axial ME-GRE images."

We also highlight a promising avenue of defining neuroanatomical landmarks based on the shape rather than the size of the gray matter:

Section 4.1: "We anticipate that, for in vivo studies, the shape of GM would serve as an even better neuroanatomical landmark than its size."

4. Recommendations for Clinical Practice: Lastly, providing clear recommendations for clinical practice based on the study’s findings, especially in terms of improving MRI accuracy in spinal cord assessments, would enhance its applicability.

In our previous study (Büeler et al., 2022; reference #17), we provided recommendations on the optimal settings to achieve precise cross-sectional area measurements for the spinal cord and gray matter, with direct clinical implications. The present paper extends our research by providing recommendations for landmark definition and adjusting for the length of the conus medullaris, which are primarily beneficial for group-level analyses but less impactful for individual subjects.

An important recommendation we provide is that all images be segmented by the same rater:

Section 4.3: "Notably, the inter-rater CV values for CSA measurements were approximately twice as high as the corresponding intra-rater CV values. Consequently, we strongly advocate for the segmentation of all images by the same rater."

---

## [Decision Letter · Decision Letter 1]

17 Mar 2024

Improved inter-subject alignment of the lumbosacral cord for group-level in vivo gray and white matter assessments: A scan-rescan MRI study at 3T

PONE-D-23-37329R1

Dear Dr. David,

We’re pleased to inform you that your manuscript has been judged scientifically suitable for publication and will be formally accepted for publication once it meets all outstanding technical requirements.

Kind regards,

Ramada Rateb Khasawneh

Academic Editor

PLOS ONE

Additional Editor Comments (optional):

IT IS A GOOD ARTICEL NOW .. GOOD LUCK

Reviewers' comments:

Reviewer's Responses to Questions

**Comments to the Author**

1. If the authors have adequately addressed your comments raised in a previous round of review and you feel that this manuscript is now acceptable for publication, you may indicate that here to bypass the “Comments to the Author” section, enter your conflict of interest statement in the “Confidential to Editor” section, and submit your "Accept" recommendation.

Reviewer #1: All comments have been addressed

Reviewer #2: All comments have been addressed

2. Is the manuscript technically sound, and do the data support the conclusions?

Reviewer #1: Yes

Reviewer #2: Yes

3. Has the statistical analysis been performed appropriately and rigorously? 

Reviewer #1: Yes

Reviewer #2: Yes

4. Have the authors made all data underlying the findings in their manuscript fully available?

Reviewer #1: Yes

Reviewer #2: Yes

5. Is the manuscript presented in an intelligible fashion and written in standard English?

Reviewer #1: Yes

Reviewer #2: Yes

6. Review Comments to the Author

Reviewer #1: Thank you for the submission of your revised manuscript, which I feel is now improved especially concerning clarifications in the methods section. I look forward to your upcoming works in the field of spinal cord imaging.

Reviewer #2: Dear Authors,

I have reviewed the revisions and responses to my comments for your manuscript and appreciate your thorough efforts to address the concerns raised. I look forward to seeing your valuable work contribute to the field.

7. PLOS authors have the option to publish the peer review history of their article (what does this mean?). If published, this will include your full peer review and any attached files.

Reviewer #1: No

Reviewer #2: **Yes: **R. Heller

---

## [Editor Report · Acceptance letter]

4 Apr 2024

PONE-D-23-37329R1 

PLOS ONE

Dear Dr. David, 

I'm pleased to inform you that your manuscript has been deemed suitable for publication in PLOS ONE. Congratulations! Your manuscript is now being handed over to our production team.

Kind regards, 

on behalf of

Dr. Ramada Rateb Khasawneh 

Academic Editor

PLOS ONE